# Variational Inference for SDEs Driven by Fractional Noise

**Rembert Daems** [1,2]    **Manfred Opper** [3,4,5]    **Guillaume Crevecoeur** [1,2]    **Tolga Birdal** [6]

[1] D2LAB, Ghent University, Belgium

[2] MIRO core lab, Flanders Make@UGent, Belgium

[3] Dept. of Theor. Comp. Science, Technical University of Berlin, Germany

[4] Inst. of Mathematics, University of Potsdam, Germany

[5] Centre for Systems Modelling and Quant. Biomed., University of Birmingham, UK

[6] Dept. of Computing, Imperial College London, UK

## Abstract

We present a novel variational framework for performing inference in (neural) stochastic differential equations (SDEs) driven by Markov-approximate fractional Brownian motion (fBM). SDEs offer a versatile tool for modeling real-world continuous-time dynamic systems with inherent noise and randomness. Combining SDEs with the powerful inference capabilities of variational methods, enables the learning of representative function distributions through stochastic gradient descent. However, conventional SDEs typically assume the underlying noise to follow a Brownian motion (BM), which hinders their ability to capture long-term dependencies. In contrast, fractional Brownian motion (fBM) extends BM to encompass non-Markovian dynamics, but existing methods for inferring fBM parameters are either computationally demanding or statistically inefficient. In this paper, building upon the Markov approximation of fBM, we derive the evidence lower bound essential for efficient variational inference of posterior path measures, drawing from the well-established field of stochastic analysis. Additionally, we provide a closed-form expression to determine optimal approximation coefficients. Furthermore, we propose the use of neural networks to learn the drift, diffusion and control terms within our variational posterior, leading to the variational training of neural-SDEs. In this framework, we also optimize the Hurst index, governing the nature of our fractional noise. Beyond validation on synthetic data, we contribute a novel architecture for variational latent video prediction,—an approach that, to the best of our knowledge, enables the first variational neural-SDE application to video perception.

## 1 Introduction

Our surroundings constantly evolve over time, influenced by several dynamic factors, manifesting in various forms, from the weather patterns and the ebb & flow of financial markets to the movements of objects (Yu et al., 2023; Rempe et al., 2021) & observers, and the subtle deformations that reshape our environments (Gojcic et al., 2021). Stochastic differential equations (SDEs) provide a natural way to capture the randomness and continuous-time dynamics inherent in these real-world processes. To extract meaningful information about the underlying system, *i.e.* to infer the model parameters and to accurately predict the unobserved paths, variational inference (VI) (Bishop & Nasrabadi, 2006) is used as an efficient means, computing the posterior probability measure over paths (Opper, 2019; Li et al., 2020; Ryder et al., 2018)[1].

The traditional application of SDEs assumes that the underlying noise processes are generated by standard Brownian motion (BM) with independent increments. Unfortunately, for many practical scenarios, BM falls short of capturing the full complexity and richness of the observed real data,

---

[1]KL divergence between two SDEs over a finite time horizon has been well-explored in the control literature (Theodorou, 2015; Kappen & Ruiz, 2016).

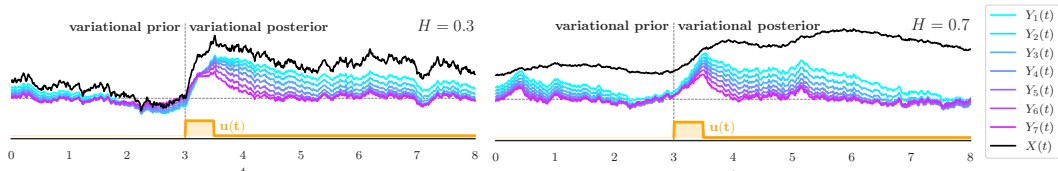

Figure 1: We leverage the Markov approximation, where the non-Markovian fractional Brownian motion with Hurst index $H$ is approximated by a linear combination of a finite number of Markov processes $(Y_1(t), \ldots, Y_K(t))$, and propose a variational inference framework in which the posterior is steered by a control term $u(t)$. Note the long-term memory behaviour of the processes, where individual $Y_k(t)$s have varying transient effects, from $Y_1(t)$ having the longest memory to $Y_7(t)$ the shortest, and tend to forget the action of $u(t)$ after a certain time frame.

which often contains long-range dependencies, rare events, and intricate temporal structures that cannot be faithfully represented by a Markovian process. The non-Markovian fractional Brownian motion (fBM) (Mandelbrot & Van Ness, 1968) extends BM to stationary increments with a more complex dependence structure, *i.e.* long-range dependence vs. roughness/regularity controlled by its *Hurst index* (Gatheral et al., 2018). Yet, despite its desirable properties, the computational challenges and intractability of analytically working with fBMs pose significant challenges for inference.

In this paper, we begin by providing a tractable variational inference framework for SDEs driven by fractional Brownian motion (Types I & II). To this end, we benefit from the relatively under-explored *Markov representation* of fBM and path-wise approximate fBM through a linear combination of a finite number of Ornstein–Uhlenbeck (OU) processes driven by a common noise (Carmona & Coutin, 1998a;b; Harms & Stefanovits, 2019). We further introduce a differentiable method to optimise for the associated coefficients and conjecture (as well as empirically validate) that this *strong* approximation enjoys super-polynomial convergence rates, allowing us to use a handful of processes even in complex problems.

Such *Markov-isation* also allows us to inherit the well-established tools of traditional SDEs including Girsanov's change of measure theorem (Øksendal & Øksendal, 2003), which we use to derive and maximise the corresponding *evidence lower bound* (ELBO) to yield posterior path measures as well as maximum likelihood estimates as illustrated in Fig. 1. We then use our framework in conjunction with neural networks to devise VI for neural-SDEs (Liu et al., 2019; Li et al., 2020) driven by the said fractional diffusion. We deploy this model along with a novel neural architecture for the task of enhanced video prediction. To the best of our knowledge, this is the first time either fractional or variational neural-SDEs are used to model videos. Our contributions are:

- We make accessible the relatively uncharted Markovian embedding of the fBM and its strong approximation, to the machine learning community. This allows us to employ the traditional machinery of SDEs in working with non-Markovian systems.
- We show how to balance the contribution of Markov processes by optimising for the combination coefficients in closed form. We further estimate the (time-dependent) Hurst index from data.
- We derive the evidence lower bound for SDEs driven by approximate fBM of both Types I and II.
- We model the drift, diffusion and control terms in our framework by neural networks, and propose a novel architecture for video prediction.

We make our implementation publicly available under: github.com/VideoNeuralSDE/MAFBM.

## 2 RELATED WORK

**Fractional noises and neural-SDEs**. fBM (Mandelbrot & Van Ness, 1968) was originally used for the simulation of rough volatility in finance (Gatheral et al., 2018). Using the Lemarié-Meyer wavelet representation, Allouche et al. (2022) provided a large probability bound on the deep-feedforward RELU network approximation of fBM, where up to log terms, a uniform error of $O(N^{-H})$ is achievable with $\log(N)$ hidden layers and $O(N)$ parameters. Tong et al. (2022) approximated the fBM (only Type II) with sparse Gaussian processes. Unfortunately, they are limited to Euler-integration and to the case of $H > 1/3$. Their model was also not applied to videos. Recently, Yang et al. (2023) applied Levy driven neural-SDEs to times series prediction and Hayashi & Nakagawa (2022) considered neural-SDEs driven by fractional noise. Neither of those introduce a variational framework. Both Liao et al. (2019); Morrill et al. (2021) worked with *rough path theory*

to model long time series via rough neural-SDEs. To the best of our knowledge, we are the firsts to devise a VI framework for neural-SDEs driven by a path-wise (strong) approximation of fBM.

**SDEs and visual understanding**. Apart from the recent video diffusion models (Luo et al., 2023; Yang et al., 2022; Ho et al., 2022), SDEs for spatiotemporal visual generation is relatively unexplored. Park et al. (2021); Ali et al. (2023) used neural-ODEs to generate and manipulate videos while (Rempe et al., 2020) used neural-ODEs for temporal 3D point cloud modeling. SDENet (Kong et al., 2020) and MDSDE-Net (Zhang et al., 2023) learned drift and diffusion networks for uncertainty estimation of images using out-of-distribution data. Tong et al. (2022) used approximate-fBMs in score-based diffusion modeling for image generation. Gordon & Parde (2021) briefly evaluated different neural temporal models for video generation. While Babaeizadeh et al. (2018) used VI for video prediction, they did not employ SDEs. To the best of our knowledge, we are the firsts to use neural-SDEs in a variational framework for video understanding.

# 3 BACKGROUND

We first tailor and make accessible the fractional Brownian Motion (fBM) and its relatively less explored Markov approximations for the learning community. We then describe the SDEs driven by fBM and its approximation before delving into the inference. We leave the proofs to our appendix.

## 3.1 FRACTIONAL BROWNIAN MOTION (FBM) & ITS MARKOV APPROXIMATION

**Definition 1** (Fractional Brownian Motion (Types I & II)). *fBM is a self-similar, non-Markovian, non-martingale, zero-mean Gaussian process* $(B_H(t))_{t \in [0,T]}$ *for $T > 0$ with a covariance of either*

$$\mathbb{E}\left[B_H^{(I)}(t)B_H^{(I)}(s)\right] = \frac{1}{2}(|t|^{2H} + |s|^{2H} - |t-s|^{2H}) \qquad \text{(Type I)} \quad (1)$$

$$\mathbb{E}\left[B_H^{(II)}(t)B_H^{(II)}(s)\right] = \frac{1}{\Gamma^2(H+1/2)}\int_0^s ((t-u)(s-u))^{H-1/2}\,\mathrm{d}u \qquad \text{(Type II)} \quad (2)$$

*where $t > s$, $0 < H < 1$ is the* Hurst index, *superscripts denote the types and $\Gamma$ is the Gamma function.*

fBM recovers *Brownian motion* (BM) for $H = 1/2$ (regular diffusion) and generalizes it for other choices. The increments are (i) positively correlated for $H > 1/2$ (super-diffusion) where the tail behaviour is infinitely heavier than that of BM, and (ii) negatively correlated for $H < 1/2$ (sub-diffusion), with variance $\mathbb{E}\left(|B_H^{(I)}(t) - B_H^{(I)}(s)|^2\right) = |t-s|^{2H}$ for Type I. The Type II model implies nonstationary increments of which the marginal distributions are dependent on the time relative to the start of the observed sample, *i.e.* all realizations would have to be found very close to the unconditional mean (*i.e.* , the origin) (Lim & Sithi, 1995; Davidson & Hashimzade, 2009).

**Definition 2** (Integral representations of fBM). $B_H^{(I,II)}$ *admit the following integral forms due to the Mandelbrot van-Ness and Weyl representations, respectively (Mandelbrot & Van Ness, 1968):*

$$B_H^{(I)}(t) = \frac{1}{\Gamma(H+1/2)}\int_{-\infty}^t \left[K^{(I)}(t,s) := \left((t-s)^{H-1/2} - (-s)_+^{H-1/2}\right)\right]\mathrm{d}W(s) \qquad (3)$$

$$= \frac{1}{\Gamma(H+1/2)}\left(\int_{-\infty}^0 \left((t-s)^{H-1/2} - (-s)^{H-1/2}\right)\mathrm{d}W(s) + \int_0^t (t-s)^{H-1/2}\,\mathrm{d}W(s)\right)$$

$$B_H^{(II)}(t) = \frac{1}{\Gamma(H+1/2)}\int_0^t \left[K^{(II)}(t,s) := (t-s)^{H-1/2}\right]\mathrm{d}W(s) \qquad (4)$$

where $K^{(I)}$ and $K^{(II)}$ are the kernels corresponding to Types I and II, respectively.

**Proposition 1** (Markov representation of fBM (Harms & Stefanovits, 2019)). *The long memory processes $B_H^{(I,II)}(t)$ can be represented by an infinite linear combination of Markov processes, all driven by the same Wiener noise, but with different time scales, defined by speed of mean reversion $\gamma$. For both types we have representations of the form:*

$$B_H(t) = \begin{cases} \int_0^\infty (Y_\gamma(t) - Y_\gamma(0))\mu(\gamma)\,\mathrm{d}\gamma, & H < 1/2, \\ -\int_0^\infty \partial_\gamma(Y_\gamma(t) - Y_\gamma(0))\nu(\gamma)\,\mathrm{d}\gamma, & H > 1/2 \end{cases}, \qquad (5)$$

where $\mu(\gamma) = \gamma^{-(H+1/2)}/\left(\Gamma(H+1/2)\Gamma(1/2-H)\right)$ and $\nu(\gamma) = \gamma^{-(H-1/2)}/(\Gamma(H+1/2)$ $\Gamma(3/2-H))$. *Note, these non–negative densities are not normalisable. To simplify notation, we will drop explicit dependency on the types $(I, II)$ in what follows. For each $\gamma \geq 0$, and for both types $I$ and $II$, the processes $Y_\gamma(t)$ are OU processes which are solutions to the SDE $dY_\gamma(t) = -\gamma Y_\gamma(t)\,dt + dW(t)$. This SDE is solved by*

$$Y_\gamma(t) = Y_\gamma(0)e^{-\gamma t} + \int_0^t e^{-\gamma(t-s)}\,dW(s). \tag{6}$$

*"Type I" and "Type II" differ in the initial conditions $Y_\gamma(0)$. One can show that:*

$$Y_\gamma^{(I)}(0) = \int_{-\infty}^0 e^{\gamma s}\,dW(s) \qquad \text{and} \qquad Y_\gamma^{(II)}(0) = 0. \tag{7}$$

**Definition 3** (Markov approximation of fBM (MA-fBM))**.** *Eq. (5) suggests that $B_H(t)$ could be well approximated by a Markov process $\hat{B}_H(t)$ by (i) truncating the integrals at finite $\gamma$ values $(\gamma_1...\gamma_K)$ and (ii) approximating the integral by a numerical quadrature as a finite linear combination involving quadrature points and weights $\{\omega_k\}$. Changing the notation $Y_{\gamma_k}(t) \to Y_k(t)$:*

$$B_H(t) \approx \hat{B}_H(t) \equiv \sum_{k=1}^K \omega_k \left(Y_k(t) - Y_k(0)\right), \tag{8}$$

*where for fixed $\gamma_k$ the choice of $\omega_k$ depends on $H$ and the choice of "Type I" or "Type II". For "Type II", we set $Y_k(0) = 0$. Since $Y_k(t)$ is normally distributed (Harms & Stefanovits, 2019, Thm. 2.16) and can be assumed stationary for "Type I", we can simply sample $\left(Y_1^{(I}(0), \ldots, Y_K^{(I)}(0)\right)$ from a normal distribution with mean $\mathbf{0}$ and covariance $\boldsymbol{C}_{i,j} = 1/(\gamma_i + \gamma_j)$ (see Eq. (28)).*

This strong approximation provably bounds the sample paths:

**Theorem 1** (Alfonsi & Kebaier (2021))**.** *For rough kernels ($H < 1/2$) and $\{\omega_k\}$ following a Gaussian quadrature rule, there exists a constant $c$ per every $t \in (0, T)$ such that:*

$$\mathbb{E}|B_H^{(II)}(t) - \hat{B}_H^{(II)}(t)| \leq O(K^{-cH}), \quad \text{where} \quad 1 < c \leq 2, \tag{9}$$

*as $K \to \infty$. Note that, in our setting, $B_H^{(II)}(0) = \hat{B}_H^{(II)}(0) = 0$.*

In the literature, different choices of $\gamma_k$ and $\omega_k$ have been proposed (Harms & Stefanovits, 2019; Carmona & Coutin, 1998a; Carmona et al., 2000) and for certain choices, it is possible to obtain a superpolynomial rate, as shown by Bayer & Breneis (2023) for the Type II case. As we will show in Sec. 4.1, choosing $\gamma_k = r^{k-n}, k = 1, \ldots, K$ with $n = (K+1)/2$ (Carmona & Coutin, 1998a), we will optimise $\{\omega_k\}_k$ for both types, to get optimal rates.

## 3.2 SDEs driven by (fractional) BM

**Definition 4** (SDEs driven by BM (BMSDE))**.** *A common generative model for stochastic dynamical systems considers a set of observational data $\mathcal{D} = \{O_1, \ldots, O_N\}$, where the $O_i$ are generated (conditionally) independent at random at discrete times $t_i$ with a likelihood $p_\theta(O_i \mid X(t_i))$. The prior information about the unobserved path $\{X(t); t \in [0, T]\}$ of the latent process $X(t) \in \mathbb{R}^M$ is given by the assumption that $X(t)$ fulfils the SDE:*

$$dX(t) = b_\theta(X(t), t)\,dt + \sigma_\theta(X(t), t)\,dW(t) \tag{10}$$

*The drift function $b_\theta(X, t) \in \mathbb{R}^D$ models the deterministic part of the change $dX(t)$ of the state variable $X(t)$ during the infinitesimal time interval $dt$, whereas the diffusion matrix $\sigma_\theta(X(t), t) \in \mathbb{R}^{D \times D}$ (assumed to be symmetric and non–singular, for simplicity) encodes the strength of the added Gaussian white noise process, where $dW(t) \in \mathbb{R}^D$ is the infinitesimal increment of a vector of independent Wiener processes during $dt$.*

**Definition 5** (SDEs driven by fBM (fBMSDE))**.** *Dfn. 4 can be formally extended to the case of fractional Brownian motion replacing $dW(t)$ by $dB_H(t)$ (Guerra & Nualart, 2008):*

$$dX(t) = b_\theta(X(t), t)\,dt + \sigma_\theta(X(t), t)\,dB_H(t). \tag{11}$$

**Remark 1.** *Care must be taken in a proper definition of the diffusion part in the fBMSDE Eq. (11) and in developing appropriate numerical integrators for simulations, when the diffusion $\sigma_\theta(X(t), t)$ explicitly depends on the state $X(t)$. Corresponding stochastic integrals of the Itô type cannot be applied when $H < 1/2$ and other approaches (which are generalisations of the Stratonovich SDE for $H = \frac{1}{2}$) are necessary (Lysy & Pillai, 2013).*

## 4 METHOD

Our goal is to extend variational inference (VI) Bishop & Nasrabadi (2006) to the case where the Wiener process in Eq. (10) is replaced by an fBM as in Dfn. 5. Unfortunately, the processes defined by Eq. (11) are not Markovian preventing us from resorting to the standard Girsanov change of measure approach known for "ordinary" SDE to compute KL–divergences and ELBO functionals needed for VI (Opper, 2019). While Tong et al. (2022) leverage sparse approximations for Gaussian processes, this makes $B_H$ *conditioned* on a finite but larger number of so–called *inducing variables*. We take a completely different and conceptually simple approach to VI for fBMSDE based on the exact representation of $B_H(t)$ given in Prop. 1. To this end, we first show how the *strong* Markov-approximation in Dfn. 3 can be used to approximate an SDE driven by fBM, before delving into the VI for the *Markov-Approximate fBMSDE*.

**Definition 6** (Markov-Approximate fBMSDE (MA-fBMSDE)). *Substituting the fBM, $B_H(t)$, in Dfn. 5 by the finite linear combination of OU-processes $\hat{B}_H(t)$, we define MA-fBMSDE as:*

$$\mathrm{d}X(t) = b_\theta(X(t), t)\,\mathrm{d}t + \sigma_\theta(X(t), t)\,\mathrm{d}\hat{B}_H(t), \tag{12}$$

*where $\mathrm{d}\hat{B}_H(t) = \sum_{k=1}^K \omega_k\,\mathrm{d}Y_k(t)$ with $\mathrm{d}Y_k(t) = -\gamma_k Y_k(t)\,\mathrm{d}t + \mathrm{d}W(t)$ (cf. Dfn. 3).*

**Proposition 2.** *$X(t)$ can be augmented by the finite number of Markov processes $Y_k(t)$ (approximating $B_H(t)$) to a higher dimensional state variable of the form $Z(t) \doteq (X(t), Y_1(t), \ldots Y_K(t)) \in \mathbb{R}^{D(K+1)}$, such that the joint process of the augmented system becomes Markovian and can be described by an 'ordinary' SDE:*

$$\mathrm{d}Z(t) = h_\theta(Z(t), t)\,\mathrm{d}t + \Sigma_\theta(Z(t), t)\,\mathrm{d}W(t), \tag{13}$$

*where the augmented drift vector $h_\theta \in \mathbb{R}^{D \times (K+1)}$ and the diffusion matrix $\Sigma_\theta(Z, t) \in \mathbb{R}^{D(K+1) \times D}$ are given by*

$$h_\theta(Z, t) = \begin{pmatrix} b_\theta(X, t) - \sigma_\theta(X, t) \sum_k \omega_k \gamma_k Y_k \\ -\gamma_1 Y_1 \\ \cdots \\ -\gamma_K Y_K \end{pmatrix} \qquad \Sigma_\theta(Z, t) = \begin{pmatrix} \bar{\omega}\sigma_\theta(X, t) \\ \vec{1} \\ \vdots \\ \vec{1} \end{pmatrix}, \tag{14}$$

*where $\vec{1} = (1, 1, \ldots, 1)^\top \in \mathbb{R}^D$. We will refer to Eq. (13) as the variational prior.*

*Proof.* Each of the $D$ components of the vectors $Y_k$ use the same scalar weights $\omega_k \in \mathbb{R}$. Also, note that each $Y_k$ is driven by the *same* vector of Wiener processes. Hence, we obtain the system of SDEs given by

$$\mathrm{d}X(t) = b_\theta(X(t), t)\,\mathrm{d}t - \sigma_\theta(X(t), t)\sum_k \omega_k \gamma_k Y_k(t)\,\mathrm{d}t + \bar{\omega}\sigma_\theta(X(t), t)\,\mathrm{d}W(t)$$
$$\mathrm{d}Y_k(t) = -\gamma_k Y_k(t)\,\mathrm{d}t + \mathrm{d}W(t) \qquad \text{for} \qquad k = 1, \ldots, K \tag{15}$$

where $\bar{\omega} \doteq \sum_k \omega_k$. This system of equations can be collectively represented in terms of the augmented variable $Z(t) := (X(t), Y_1(t), \ldots Y_K(t)) \in \mathbb{R}^{D(K+1)}$ leading to a single SDE specified by Eqs. (13) and (14). □

Eq. (13) represents a standard SDE driven by Wiener noise allowing us to utilise the standard tools of stochastic analysis, such as the Girsanov change of measure theorem and derive the *evidence lower bounds* (ELBO) required for VI. This is what we will exactly do in the sequel.

**Proposition 3** (Controlled MA-fBMSDE). *The paths of Eq. (13) can be steered by adding a control term $u(X, Y_1, \ldots, Y_K, t) \in \mathbb{R}^D$ that depends on all variables to be optimised, to the drift $h_\theta$ resulting in the transformed SDE, a.k.a. the variational posterior:*

$$\mathrm{d}\tilde{Z}(t) = \left(h_\theta\left(\tilde{Z}(t), t\right) + \sigma_\theta(\tilde{Z}(t), t)u(\tilde{Z}(t), t)\right)\mathrm{d}t + \Sigma_\theta\left(\tilde{Z}(t), t\right)\mathrm{d}W(t) \tag{16}$$

*Sketch of the proof.* Using the fact that the posterior probability measure over paths $\tilde{Z}(t)$ $\{\tilde{Z}(t); t \in [0, T]\}$ is absolutely continuous w.r.t. the prior process, we apply the Girsanov theorem (*cf.* App. B.1) on Eq. (13) to write the new drift, from which the posterior SDE in Eq. (16) is obtained. □

We will refer to Eq. (16) as the *variational posterior*. In what follows, we will assume a parametric form for the control function $u(\tilde{Z}(t), t) \equiv u_\phi(\tilde{Z}(t), t)$ (as *e.g.* given by a neural network) and will devise a scheme for inferring the *variational parameters* $(\theta, \phi)$, *i.e.* variational inference.

**Proposition 4** (Variational Inference for MA-fBMSDE). *The variational parameters $\phi$ are optimised by minimising the KL–divergence between the posterior and the prior, where the corresponding evidence lower bound (ELBO) to be maximised is:*

$$\log p\left(O_1, O_2, \ldots, O_N \mid \theta\right) \geq \mathbb{E}_{\tilde{Z}_u}\left[\sum_{i=1}^{N} \log p_\theta\left(O_i \mid \tilde{Z}(t_i)\right) - \int_0^T \frac{1}{2}\left\|u_\phi\left(\tilde{Z}(t), t\right)\right\|^2 \, \mathrm{d}t\right],$$
(17)

*where the observations $\{O_i\}$ are included by likelihoods $p_\theta\left(O_i \mid \tilde{Z}(t_i)\right)$ and the expectation is taken over random paths of the approximate posterior process defined by (Eq. (16)).*

*Sketch of the proof.* Since we can use Girsanov's theorem II (Øksendal & Øksendal, 2003), the variational bound derived in Li et al. (2020) (App. 9.6.1) directly applies. □

**Remark 2.** *It is noteworthy that the measurements with their likelihoods $p_\theta\left(O_i \mid \tilde{X}(t_i)\right)$ depend only on the component $\tilde{X}(t)$ of the augmented state $\tilde{Z}(t)$. The additional variables $Y_k(t)$ which are used to model the noise in the SDE are not directly observed. However, computation of the ELBO requires initial values for all state variables $\tilde{Z}(0)$ (or their distribution). Hence, we sample $Y_k(0)$ in accordance with Dfn. 3.*

## 4.1 Optimising the approximation

We now present the details of our novel method for optimising our approximation $\hat{B}_H^{(I,II)}(t)$ for $\omega_k$. To this end, we first follow Carmona & Coutin (1998a) and choose a geometric sequence of $\gamma_k = (r^{1-n}, r^{2-n}, \ldots, r^{K-n})$, $n = \frac{K+1}{2}$, $r > 1$. Rather than relying on methods of numerical quadrature, we consider a simple measure for the quality of the approximation over a fixed time interval $[0, T]$ which can be *optimised analytically* for both types I and II.

**Proposition 5** (Optimal $\omega \doteq [\omega_1, \ldots, \omega_K]$ for $\hat{B}^{(I,II)}(t)$). *The $L_2$-error of our approximation*

$$\mathcal{E}^{(I,II)}(\omega) = \int_0^T \mathbb{E}\left[\left(\hat{B}_H^{(I,II)}(t) - B_H^{(I,II)}(t)\right)^2\right] \mathrm{d}t$$
(18)

*is minimized at $\mathbf{A}^{(I,II)}\omega = \mathbf{b}^{(I,II)}$, where*

$$\mathbf{A}_{i,j}^{(I)} = \frac{2T + \frac{e^{-\gamma_i T} - 1}{\gamma_i} + \frac{e^{-\gamma_j T} - 1}{\gamma_j}}{\gamma_i + \gamma_j}, \quad \mathbf{A}_{i,j}^{(II)} = \frac{T + \frac{e^{-(\gamma_i + \gamma_j)T} - 1}{\gamma_i + \gamma_j}}{\gamma_i + \gamma_j}$$
(19)

$$\mathbf{b}_k^{(I)} = \frac{2T}{\gamma_k^{H+1/2}} - \frac{T^{H+1/2}}{\gamma_k \Gamma(H + 3/2)} + \frac{e^{-\gamma_k T} - Q(H + 1/2, \gamma_k T)e^{\gamma_k T}}{\gamma_k^{H+3/2}}$$
(20)

$$\mathbf{b}_k^{(II)} = \frac{T}{\gamma_k^{H+1/2}} P(H + 1/2, \gamma_k T) - \frac{H + 1/2}{\gamma_k^{H+3/2}} P(H + 3/2, \gamma_k T).$$
(21)

$P(z, x) = \frac{1}{\Gamma(z)} \int_0^x t^{z-1} e^{-t} \, \mathrm{d}t$ *is the regularized lower incomplete gamma function and* $Q(z, x) = \frac{1}{\Gamma(z)} \int_x^\infty t^{z-1} e^{-t} \, \mathrm{d}t$ *is the regularized upper incomplete gamma function.*

*Sketch of the proof.* By expanding the $L_2$-error we find a tractable quadratic form of the criterion:

$$\mathcal{E}^{(I,II)}(\omega) = \int_0^T \mathbb{E}\left[\left(\hat{B}_H^{(I,II)}(t) - B_H^{(I,II)}(t)\right)^2\right] \mathrm{d}t$$
(22)

$$= \int_0^T \left(\mathbb{E}\left[\hat{B}_H^{(I,II)}(t)^2\right] + \mathbb{E}\left[B_H^{(I,II)}(t)^2\right] - 2\mathbb{E}\left[\hat{B}_H^{(I,II)}(t)B_H^{(I,II)}(t)\right]\right) \mathrm{d}t$$

$$= \omega^T \mathbf{A}^{(I,II)}\omega - 2\mathbf{b}^{(I,II)^T}\omega + \mathrm{const},$$

whose non-trivial minimum is attained as the solution to the system of equations $\mathbf{A}^{(I,II)}\omega = \mathbf{b}^{(I,II)}$. We refer the reader to App. D.2 for the full proof and derivation. □

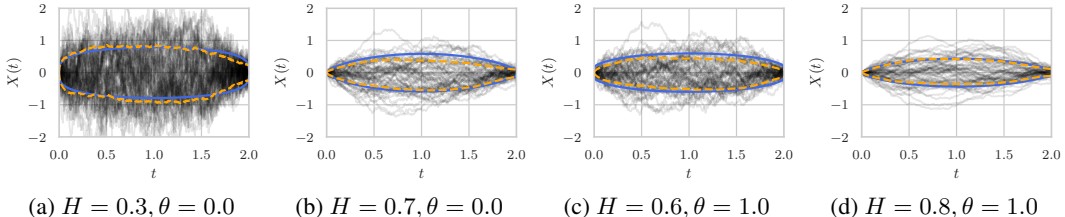

(a) $H = 0.3, \theta = 0.0$    (b) $H = 0.7, \theta = 0.0$    (c) $H = 0.6, \theta = 1.0$    (d) $H = 0.8, \theta = 1.0$

Figure 2: The true variance (blue) of a fOU bridge matches the empirical variance (dashed orange) of our trained models. The transparent black lines are the sampled approximate posterior paths used to calculate the empirical variance.

## 5 EXPERIMENTS

We implemented our method in JAX (Bradbury et al., 2018), using Diffrax (Kidger, 2021) for SDE solvers, Optax (Babuschkin et al., 2020) for optimization, Diffrax (Babuschkin et al., 2020) for distributions and Flax (Heek et al., 2023) for neural networks. Unlike Tong et al. (2022) our approach is agnostic to discretization and the choice of the solver. Hence, in all experiments we can use the *Stratonovich–Milstein* solver, *cf.* App. E for more details.

**Recovering the fractional Ornstein–Uhlenbeck bridge**. Applying our method on linear problems, allows comparing empirical results to analytical formulations derived *e.g.* using Gaussian process methodology Rasmussen et al. (2006). We begin by assessing the reconstruction capability of our method on a fractional Ornstein–Uhlenbeck (fOU) bridge, that is an OU–process driven by fBM: $dX(t) = -\theta X(t)\, dt + dB_H$, starting at $X(0) = 0$ and conditioned to end at $X(T) = 0$. Following the rules of Gaussian process regression (Rasmussen et al., 2006, Eq. 2.24), we have an analytical expression for the posterior covariance:

$$\mathbb{E}\left[\tilde{X}(t)^2\right] = K(t,t) - \begin{bmatrix} K(t,0) & K(t,T) \end{bmatrix} \begin{bmatrix} K(0,0) & K(T,0) \\ K(0,T) & K(T,T) + \sigma^2 \end{bmatrix}^{-1} \begin{bmatrix} K(0,t) \\ K(T,t) \end{bmatrix} \tag{23}$$

where $K(t,\tau)$ is the prior kernel and the observation noise is 0 for $X(0)$ and $\sigma$ for $X(T)$. If $\theta = 0$, $K(t,\tau) = \mathbb{E}\left[B_H(t)B_H(\tau)\right]$ (Eq. (1)) and if $\theta > 0$ and $H > 1/2$, the kernel admits the following form (Lysy & Pillai, 2013, Appendix A):

$$K(t,\tau) = \frac{(2H^2 - H)}{2\theta}\left(e^{-\theta|t-\tau|}\left[\frac{\Gamma(2H-1) + \Gamma(2H-1, |t-\tau|)}{\theta^{2H-1}} + \int_0^{|t-\tau|} e^{\theta u} u^{2H-2}\, du\right]\right) \tag{24}$$

where $\Gamma(z,x) = \int_x^\infty t^{z-1} e^{-t}\, dt$ is the upper incomplete Gamma function. This allows us to compare the true posterior variance with the empirical variance of a model that is trained by maximizing the ELBO. for a data point $X(T) = 0$. As this is equivalent to the analytical result (Eq. (23)), we can compare the variances over time. As plotted in Fig. 2, for various $H$ and $\theta$ values, our VI can correctly recover the posterior variance, *cf.* App. F for additional results.

**Estimating time-dependent Hurst index.** Since our method of optimizing $\omega_k$ is tractable and differentiable, we can directly optimize a parameterized $H$ by maximizing the ELBO. Also a time–dependent Hurst index $H(t)$ can be modelled, leading to multifractional Brownian Motion (Peltier & Véhel, 1995). We directly compare with a toy problem presented in (Tong et al., 2022, Sec. 5.2). We use the same model for $H(t)$, a neural network with one hidden layer of 10 neurons and activation function tanh, and a final sigmoid activation, and the same input $[\sin(t), \cos(t), t]$. We use $\hat{B}_H^{(II)}$ since their method is Type II. Fig. 3 shows a reasonable estimation of $H(t)$, which is more accurate than the result from Tong et al. (2022), *cf.* App. E for more details.

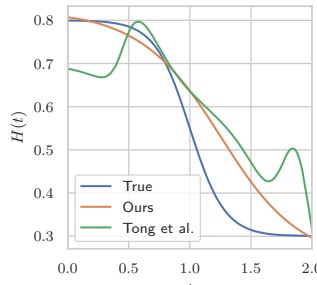

Figure 3: Estimating time–dependent $H(t)$ from data.

**Latent video models** To assess the video modelling capabilities of our framework, we train models on stochastic video datasets. The prior drift $h_\theta$, diffusion $\sigma_\theta$ and control term $u$ are parameterized

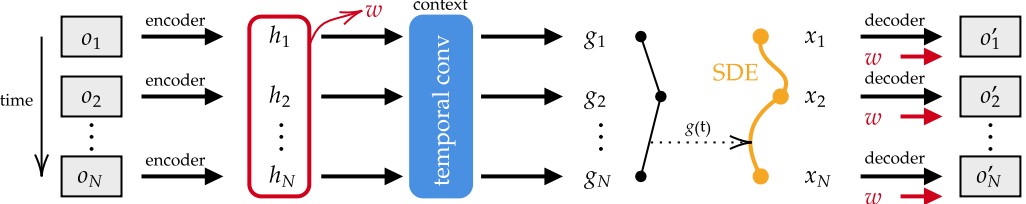

Figure 4: Schematic of the latent SDE video model. Video frames $\{o_i\}_i$ are encoded to vectors $\{h_i\}_i$. The static content vector $w$, that is free of the dynamic information, is inferred from $\{h_i\}_i$. The context model processes the information with temporal convolution layers, so that its outputs $\{g_i\}_i$ contain information from neighbouring frames. A linear interpolation on $\{g_i\}_i$ allows the posterior SDE model to receive time–appropriate information $g(t)$, at (intermediate) time–steps chosen by the SDE solver. Finally, the states $\{x_i\}_i$ and static $w$ are decoded to reconstruct frames $\{o'_i\}_i$.

by neural networks. The prior model is used as a stochastic video predictor, where we condition on the first $N$ frames to predict the next frames in the sequence. More intuitively, the posterior model reconstructs the given sequence of frames, while minimizing the control actions of $u$. This leads to a prior that will model the dataset, so that the posterior will be able to model the *specific* data sequence during training with minimal $u$ input. It is paramount that the control function $u$ receives relevant information during the SDE integration, so that it can steer the SDE in the right direction. See Fig. 4 for a schematic explanation of our model and App. E for a detailed explanation of submodel architectures and hyperparameters.

We evaluate the stochastic video predictions by sampling 100 predictions and reporting the Peak Signal-to-Noise Ratio (PSNR) of the best sample, calculated frame-wise and averaged over time. This is the same approach as Franceschi et al. (2020) which allows a direct comparison. Furthermore, we report the ELBO on the test set, indicating how well the model has captured the data.

We train models on Stochastic Moving MNIST (Denton & Fergus, 2018), a video dataset where two MNIST numbers move on a canvas and bounce off the edge with random velocity in a random direction. Our MA-fBM driven model is on par with closely related discrete-time methods such as SVG (Denton & Fergus, 2018) or SLRVP Franceschi et al. (2020), in terms of PSNR, and is better than the BM baseline in terms of PSNR and ELBO (Tab. 1).

Table 1: Stochastic Moving MNIST results.

| Model | ELBO | PSNR |
|---|---|---|
| SVG | N/A | 14.50 |
| SLRVP | N/A | 16.93 |
| BM | −913.60 | 14.90 |
| MA-fBM | −608.00 | 15.30 |

The Hurst index was optimized during training, and reached $H = 0.90$ at convergence (long-term memory), indicating that MA-fBM is better suited to the data than BM.

Table 2: Double pendulum.

| Model | ELBO | PSNR |
|---|---|---|
| BM | −545.13 | 26.11 |
| MA-fBM | −636.61 | 27.09 |

We also report results on a real-world video dataset of a double pendulum (Asseman et al., 2018), where we investigate whether the chaotic behaviour can be modelled by an SDE driven by fBM. Our MA-fBM driven model is better than the BM baseline, both for the test set ELBO as for the PSNR metric (Tab. 2). The Hurst index reached a value of $H = 0.93$ at convergence. See Fig. 5 for stochastic video predictions and App. F.3 for additional qualitative results.

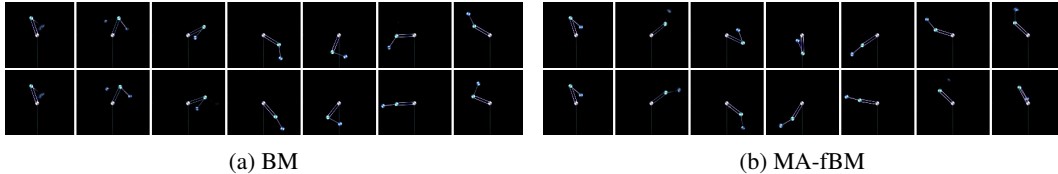

(a) BM        (b) MA-fBM

Figure 5: Stochastic video predictions using the trained prior of a model driven by BM (a) and a model driven by MA-fBM (b) trained on the double pendulum dataset. The initial state is conditioned on the same data for all samples. Two samples are shown for each model, and 7 evenly spaced frames from the total of 20 frames in the sequence are shown. The MA-fBM samples show a more diverse, chaotic behaviour, thus better capturing the dynamics in the data.

## 5.1 ABLATIONS & FURTHER STUDIES

**Numerical study of the Markov approximation.** By numerically evaluating the criterion $\mathcal{E}^{(II)}$ we can investigate the effect of $K$, the number of OU–processes, on the quality of the approximation. Fig. 6 indicates that the approximation error diminishes by increasing $K$. However, after a certain threshold the criterion saturates, depending on $H$. Adding more processes, especially for low $H$ brings diminishing returns. The rapid convergence evidenced in this empirical result well agrees with the theoretical findings of (Bayer & Breneis, 2023) especially for the *rough processes* where $H < 1/2$, as recalled in Thm. 1.

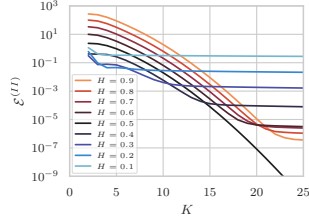

Figure 6: $\mathcal{E}^{(II)}$ vs. $K$.

**MSE of the generated trajectories for MA-fBM and for varying $K$.** On a more practical level, we take integration and numerical errors into account by simulating paths using MA-fBM and comparing to paths of the true integral driven by the *same* Wiener noise. This is only possible for Type II, as for Type I one would need to start the integration from $-\infty$. Paths are generated from $t = 0$ to $t = 10$, with 4000 integration steps for the approximation and 40000 for the true integral. We generate the paths over a range of Hurst indices and different $K$ values. For each setting, 16 paths are sampled. Our approach for optimising $\omega_k$ values (Sec. 4.1) is compared to a baseline where $\omega_k$ is derived by a piece-wise approximation of the Laplace integral (*cf.* App. D.1). Fig. 7 shows

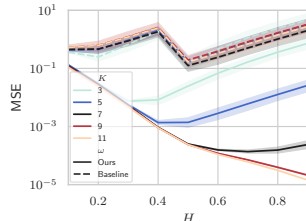

Figure 7: Mean square error (MSE) with $95\%$ confidence intervals vs. $H$ for varying $K$.

considerably better results in favor of our approach. Increasing $K$ has a rapid positive impact on the accuracy of the approximation with diminishing returns, further confirming our theoretical insights in Sec. 3. We provide examples of individual trajectories generated in this experiment in App. F.1.

**Impact of $K$ and the #parameters on inference time.** We investigate the factors that influence the training time in Fig. 8, where $K$ OU–processes are gradually included to systems with increasing number of network parameters. Note that, since our approximation is driven by 1 *Wiener process*, and the control function $u(\tilde{Z}(t), t)$ is scalar, the impact on computational load of including more processes is limited and the run-time is still dominated by the size of the neural networks. This is good news as different applications might demand different number of OU–processes.

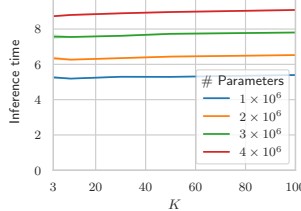

Figure 8: $K$ vs. the run-time.

## 6 CONCLUSION

In this paper, we have proposed a new approach for performing variational inference on stochastic differential equations driven by *fractional* Brownian motion (fBM). We began by uncovering the relatively unexplored Markov representation of fBM, allowing us to approximate non-Markovian paths using a linear combination of Wiener processes. This approximation enabled us to derive evidence lower bounds through Girsanov's change of measure, yielding posterior path measures as well as likelihood estimates. We also solved for optimal coefficients for combining these processes, in closed form. Our diverse experimental study, spanning fOU bridges and Hurst index estimation, have consistently validated the effectiveness of our approach. Moreover, our novel, continuous-time architecture, powered by Markov-approximate fBM driven neural-SDEs, has demonstrated improvements in video prediction, particularly when inferring the Hurst parameter during inference.

**Limitations and future work**. In our experiments, we observed increased computational overhead for larger time horizons due to SDE integration, although the expansion of the number of processes incurred minimal runtime costs. We have also observed super-polynomial convergence empirically and recalled weaker polynomial rates in the literature. Our Markov approximation still lacks a tight convergence bound. Our future work will also extend our framework to (fractional) Levy processes, which offer enhanced capabilities for modeling *heavy-tailed* noise/data distributions.

**Acknowledgments**. The authors thank Jonas Degrave and Tom Lefebvre for insightful discussions. This research received funding from the Flemish Government under the "Onderzoeksprogramma Artificiële Intelligentie (AI) Vlaanderen" programme. Furthermore it was supported by Flanders Make under the SBO project CADAIVISION. MO has been partially funded by Deutsche Forschungsgemeinschaft (DFG) - Project - ID 318763901 - SFB1294.

## ETHICS STATEMENT

Our work is driven by a dedication to the advancement of knowledge and the betterment of society. While being largely theoretical, similar to many works advancing artificial intelligence, our work deserves an ethical consideration, which we present below.

All of our experiments were either run on publicly available datasets or on data that is synthetically generated. No human or animal subjects have been involved at any stage of this work. Our models are designed to enhance the understanding and prediction of real-world processes without causing harm or perpetuating unjust biases, unless provided in the datasets. While we do not foresee any issue with methodological bias, we have not analyzed the inherent biases of our algorithm and there might be implications in applications demanding utmost fairness.

We aptly acknowledge the contributions of researchers whose work laid the foundation for our own. Proper citations and credit are given to previous studies and authors. All authors declare that there are no conflicts of interest that could compromise the impartiality and objectivity of this research. All authors have reviewed and approved the final manuscript before submission.

## REPRODUCIBILITY STATEMENT

We are committed to transparency in research and for this reason make our implementation publicly available under: github.com/VideoNeuralSDE/MAFBM. Considerable parts involve: (i) the Markov approximation and optimisation of the $\omega_k$ coefficients; (ii) maximising ELBO to perform variational inference between the prior $\mathrm{d}Z(t)$ and the posterior $\mathrm{d}\hat{Z}(t)$ and (iii) the novel neural-SDE based video prediction architecture making use of all our contributions. Our code replicates some of our evaluations for both of the datasets involved.

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

## A  FURTHER DISCUSSIONS

**Difference between a Type I and a Type II fBM.** Type I, also called Mandelbrot-Van Ness or 'standard' fBM is the most prevalent definition of fBM. Type II, also called Riemann-Liouville fBM, is historically most used in econometric literature. As can be seen in the integral definitions (Eqs. (3) and (4)), Type II omits the first integral from $-\infty$ to $0$ in the definition of Type I. So Type II is, in a sense, a simplification. Yet, to the best of our knowledge, its covariance (Eq. (2)) has no simple analytical expression. On the other hand, the Type I covariance is the straightforward, well known Eq. (1).

As described by Lim & Sithi (1995); Marinucci & Robinson (1999) the main difference is that Type I has stationary increments, and Type II has non-stationary increments. This means that Type II has a larger emphasis on the origin $t = 0$, which might not be favourable for some applications. For example, if during training we sample a sequence from a video dataset at a random start point, this $t = 0$ has no special or distinguished meaning and should not be treated differently by the driving fBM process. In other words, Type I ensures a shift in time has no effect on its increments. However, this is not the case for Type II. This difference is relevant for our framework, since the increments are driving the SDE.

**Optimal choices for $\omega$ and $\gamma$ values.** Regarding the Type II case, there are different ways of determining $\gamma_k$ and $\omega_k$ in the literature (Carmona & Coutin, 1998a; Bayer & Breneis, 2023; Harms & Stefanovits, 2019) some of which can lead to super-polynomial convergence (Bayer & Breneis, 2023) under certain assumptions, while more general choices are still shown to converge, though with a weaker rate (Alfonsi & Kebaier, 2021) while still being *strong* (path-wise) and of arbitrarily high polynomial order (Harms, 2020). Some of these works state that such geometric choice of the quadrature intervals simplifies the proofs while being not optimal and smarter choices can exist (even with better rate of convergence). This is the reason why we believe that our computationally tractable, closed form expressions which optimally solve for these values lead to good, super-polynomial convergence both for types II and I (since the first type also admits a similar type of analysis).

**Practical considerations for choosing $\gamma_k$.** Defining $\gamma_k$ as $(1/\gamma_{\max}, \ldots, \gamma_{\max})$ is a convenient way to indicate some practical considerations for choosing $\gamma_k$. Carmona & Coutin (1998b) show that $\gamma \, \mathrm{d}t > 1/2$ leads to unstable integration of the OU–process, where $\mathrm{d}t$ is the integration step. Care should be taken that $\gamma_{\max} \, \mathrm{d}t < 1/2$, either by decreasing $\gamma_{\max}$ or decreasing the integration step $\mathrm{d}t$. Additionally, choosing large values for $\gamma$ is undesirable for numerical reasons. Especially when using lower precision, numerical overflow can be a problem. Since an OU–process reaches equilibrium after time $1/\gamma$, a practical lower bound for $\gamma_{\max}$ is the length of the modelled sequences. This ensures that memory of the MA-fBM process is modelled for at least the length of the sequence.

**Time horizon for optimising $\omega_k$.** The closed form expressions for $\omega_k$ are in function of $H$ and the time horizon $T$ (Prop. 5). Since the criterion is defined over the time interval $[0, \ T]$, it makes sense to choose $T$ equal to the typical (or maximal) length of sequences in the modelled dataset. Specifically for "Type I", we advise to choose $T$ at two or three times the modelled sequence length, as at $t = 0$, this process is already at equilibrium, and its 'history' should be accounted for in the criterion. We have observed better empirical results when choosing $T$ at a multiple of the sequence length.

**Further clarification on the distinction with Tong et al. (2022).** Our work mainly differs with Tong et al. (2022) in two ways: (i) fractional Brownian motion (fBM) is approximated as a a Gaussian process (GP), (ii) only the Type II representation of fBM is used as as an integral over increments of the Wiener process. Tong et al. (2022) perform a finite time discretization of the Type II integral to obtain a first approximation of the increments of fBM. In a second step, this approximate GP is further approximated using a sparse GP approach based on a smaller set of pseudo or inducing points which are distributed over time. Conditioned on the inducing points, samples from the sparse GP are independent random variables at each discrete time point. Finally, this (conditioned) white noise process is further interpreted in terms of the Euler discretization of an ordinary

SDE leading to effective drift and diffusions. For the latter SDE, one can apply Girsanov's theorem and the corresponding ELBO (conditioned on the inducing points) to perform inference.

Note, that their current derivation of effective drift and diffusion relies on the Euler discretization of SDE. For higher order SDE solvers, the approximation has to be adapted, which requires new derivations. As a main difference, in our paper, the approximation is not based on the discretization in the time domain but of the discretization of an integral representation (Prop. 1) over a spectrum of decay constants of Ornstein-Uhlenbeck (OU) processes (driven by the same Wiener noise). Since each OU process already represents a noise process with temporal correlations, we can expect that a linear combination of a small number of such processes can yield a good approximation of the covariance of fBM over some given time interval. Our approximation leads to a system of SDEs (without conditioning) for which the ELBO can be easily obtained. Since the time discretization of the resulting SDE is performed *after* the OU approximation, any SDE solver can be directly applied. With this flexibility, in our paper, we have chosen the second order Stratonovich–Milstein solver.

**State dependent diffusions**. For the case, where the diffusion $\sigma(X, t)$ explicitly depends on the state variable $X$, our Markovian approximation results in a 'standard' white noise SDE for the augmented system. As such, it does not suffer from problems with proper definitions of stochastic integrals as compared to the original SDE driven by fBM for such cases. Hence, a straightforward Itô–interpretation of our augmented SDE is, in principle, possible. This might indicate, at first glance, that simple numerical solvers such as Euler's method could be sufficient for simulating the augmented SDE required for computing posterior expectations for the ELBO. While this point needs further theoretical investigation, preliminary simulations for for simple models with state dependent diffusions indicate that an Euler approximation (in accordance with known results for direct simulations of SDE driven by fBM (Lysy & Pillai, 2013)) quickly lead to deviations from known analytical results. Hence, for state dependent diffusions, we resort to the Stratonovich interpretation of the augmented system and use corresponding higher order solvers Kidger (2021)[2]. This approach yields excellent (pathwise) agreements with exact analytical results as we show in Sec. 5. Although the ELBO for SDE is derived from Girsanov's change of measure theorem for Itô–SDE, by the known correspondence (resulting in a change of drift functions, when diffusions are state dependent) (Gardiner et al., 1985) between Itô and Stratonovich SDE we conclude that within this approach, optimisation of the ELBO with respect to model parameters will also yield the corresponding estimates for the Stratonovich interpretation.

**On initial values for "Type I"**. The initial values for "Type I" can be understood as resulting from an OU–process which was started at some negative time $t \to -\infty$ so that

$$Y_k^{(I)}(0) = \int_{-\infty}^{0} e^{\gamma_k s} \, \mathrm{d}W(s) \tag{25}$$

and $Y_k^{(I)}(0)$ can be considered as samples from the joint stationary distribution. Because the stationary distribution is normal (Harms & Stefanovits, 2019, Theorem 2.16) we can simply sample initial states of the $Y_k(t)$ processes for Type I with covariance $\mathbb{E}\left[Y_i(0)Y_j(0)\right]$. Using Itô isometry (Øksendal & Øksendal, 2003):

$$\mathbb{E}\left[Y_i(0)Y_j(0)\right] = \mathbb{E}\left[\int_{-\infty}^{0} e^{\gamma_i s} \, \mathrm{d}W(s) \int_{-\infty}^{0} e^{\gamma_j s} \, \mathrm{d}W(s)\right] \tag{26}$$

$$= \int_{-\infty}^{0} e^{(\gamma_i + \gamma_j)s} \, \mathrm{d}s \tag{27}$$

$$= \frac{1}{\gamma_i + \gamma_j} \,. \tag{28}$$

---

[2]see *e.g.* https://docs.kidger.site/diffrax/usage/how-to-choose-a-solver/#stochastic-differential-equations

# B   PROOFS AND FURTHER THEORETICAL DETAILS

## B.1   THE GIRSANOV THEOREM II AND THE KL DIVERGENCE OF MEASURES

We now state the variation II of the Girsanov theorem (Øksendal & Øksendal, 2003) in our notation. Let $X(t) \in \mathbb{R}^n$ be an Itô process w.r.t. measure $P$ of the form:

$$dX(t) = b_\theta (X(t), t) \, dt + \sigma_\theta (X(t), t) \, dW(t), \tag{29}$$

where $0 \le t \le T$, $W(t) \in \mathbb{R}^m$, $b_\theta (X(t), t) \in \mathbb{R}^n$ and $\sigma_\theta (X(t), t) \in \mathbb{R}^{n \times m}$. Define a measure $Q$ via:

$$\frac{dQ}{dP} = M_T := \exp \left[ - \int_0^T u(X(t), t) \, dW(t) - \frac{1}{2} \int_0^T u^2(X(t), t) \, dt \right]. \tag{30}$$

Then

$$W'(t) := \int_0^T u(X(t), t) \, dt + W(T) \tag{31}$$

is a Brownian motion w.r.t. $Q$ and the process $X(t)$ has the following representation in terms of $B'(t)$:

$$dX(t) = \alpha_\theta (X(t), t) \, dt + \sigma_\theta (X(t), t) \, dW'(t), \tag{32}$$

where the new drift is:

$$\alpha_\theta (X(t), t) = b_\theta (X(t), t) - \sigma_\theta (X(t), t) \, u (X(t), t). \tag{33}$$

We can also rewrite the Radon–Nykodim derivative in Eq. (30) as

$$\frac{dQ}{dP} = \exp \left[ \int_0^T u (X(t), t) \, dW(t) - \frac{1}{2} \int_0^T u^2 (X(t), t) \, dt \right] \tag{34}$$

$$= \exp \left[ \int_0^T u (X(t), t) \, (dW'(t) + u (X(t), t) \, dt) - \frac{1}{2} \int_0^T u (X(t), t) \, dt \right] \tag{35}$$

$$= \exp \left[ \int_0^T u (X(t), t) \, dW'(t) + \frac{1}{2} \int_0^T u (X(t), t) \, dt \right]. \tag{36}$$

Thus, similar to Li et al. (2020), we get the KL divergence

$$E_Q \left[ \ln \frac{dQ}{dP} \right] = \frac{1}{2} \int_0^T E_Q[u^2 (X(t), t)] \, dt. \tag{37}$$

# C   COVARIANCES

The full derivation of covariances between some processes relevant to this work are described here.

**Fractional Brownian motion (Type II)**. Using Itô isometry (Øksendal & Øksendal, 2003) we know that for $t > s$

$$\mathbb{E} \left[ \int_0^t (t - u)^{H-1/2} dW_u \int_0^s (s - u)^{H-1/2} dW_u \right] = \int_0^s ((t - u)(s - u))^{H-1/2} \, du \tag{38}$$

Thus

$$\mathbb{E} \left[ B_H^{(II)}(t) B_H^{(II)}(s) \right] = \frac{1}{\Gamma^2(H + 1/2)} \int_0^s ((t - u)(s - u))^{H-1/2} \, du \tag{39}$$

**OU–processes driven by the same Wiener process**. Observe two Ornstein–Uhlenbeck processes driven by the same Wiener process:

$$\begin{cases} dY_i(t) = -\gamma_i Y_i(t) \, dt + dW(t) \\ dY_j(t) = -\gamma_j Y_j(t) \, dt + dW(t) \end{cases} \tag{40}$$

Their covariance can be written as:

$$\text{Cov}(Y_i(t), Y_j(t)) = \mathbb{E}\left[(Y_i(t) - \mathbb{E}\left[Y_i(t)\right])(Y_j(t) - \mathbb{E}\left[Y_j(t)\right])\right] \tag{41}$$

$$= \mathbb{E}\left[Y_i(t)Y_j(t)\right] \tag{42}$$

$$= \mathbb{E}\left[\int_0^t e^{-\gamma_i(t-s)}\,\mathrm{d}W(s) \int_0^t e^{-\gamma_j(t-s)}\,\mathrm{d}W(s)\right] \tag{43}$$

$$= \int_0^t e^{-(\gamma_i+\gamma_j)(t-s)}\,\mathrm{d}s \tag{44}$$

$$= \frac{1}{\gamma_i+\gamma_j} - \frac{e^{-(\gamma_i+\gamma_j)t}}{\gamma_i+\gamma_j} \tag{45}$$

where Eq. (44) is obtained following the Itô isometry (Øksendal & Øksendal, 2003).

**Markov approximated fractional Brownian motion (Type I).** Recall that (Dfn. 3)

$$\hat{B}_H^{(I)}(t) = \sum_k \omega_k(Y_k(t) - Y_k(0))$$

where (Eq. (6))

$$Y_k(t) - Y_k(0) = Y_k(0)(e^{-\gamma_k t} - 1) + \int_0^t e^{-\gamma_k(t-s)}\,\mathrm{d}W(s)$$

and $\mathbb{E}[Y_i(0)Y_j(0)] = \frac{1}{\gamma_i+\gamma_j}$ (Eq. (28)). For $t > \tau$:

$$\mathbb{E}\left[\hat{B}_H^{(I)}(t)\hat{B}_H^{(I)}(\tau)\right] = \mathbb{E}\left[\left(\sum_k \omega_k\,(Y_k(t) - Y_k(0))\right)\left(\sum_k \omega_k\,(Y_k(\tau) - Y_k(0))\right)\right] \tag{46}$$

$$= \sum_{i,j} \omega_i\omega_j \mathbb{E}[(Y_i(t) - Y_i(0))\,(Y_j(\tau) - Y_j(0))] \tag{47}$$

$$= \sum_{i,j} \omega_i\omega_j \mathbb{E}\left[\left(Y_i(0)(e^{-\gamma_i t} - 1) + \int_0^t e^{-\gamma_i(t-s)}\,\mathrm{d}W(s)\right)\right. \tag{48}$$

$$\left. \cdot \left(Y_j(0)(e^{-\gamma_j \tau} - 1) + \int_0^\tau e^{-\gamma_j(\tau-s)}\,\mathrm{d}W(s)\right)\right]$$

$$= \sum_{i,j} \omega_i\omega_j \left(\mathbb{E}\left[Y_i(0)Y_j(0)\right](e^{-\gamma_i t} - 1)(e^{-\gamma_j \tau} - 1)\right. \tag{49}$$

$$\left. + \int_0^\tau (e^{-\gamma_i(t-s)}e^{-\gamma_j(\tau-s)}\,\mathrm{d}s\right)$$

$$= \sum_{i,j} \omega_i\omega_j \frac{1 - e^{-\gamma_i t} - e^{-\gamma_j \tau} + e^{-\gamma_i(t-\tau)}}{\gamma_i+\gamma_j} \tag{50}$$

**Markov approximated fractional Brownian motion (Type II).** Recall that (Dfn. 3)

$$\hat{B}_H^{(II)}(t) = \sum_k \omega_k Y_k(t), \qquad Y_k(0) = 0, \quad k = 1, \dots, K$$

and for $t > \tau$:

$$\mathbb{E}\left[\hat{B}_H^{(II)}(t)\hat{B}_H^{(II)}(\tau)\right] = \mathbb{E}\left[\left(\sum_k \omega_k Y_k(t)\right)\left(\sum_{k=1}^K \omega_k Y_k(\tau)\right)\right] \tag{51}$$

$$= \sum_{i,j} \omega_i \omega_j \mathbb{E}[Y_i(t)Y_j(\tau)] \tag{52}$$

$$= \sum_{i,j} \omega_i \omega_j \mathbb{E}\left[\int_0^t e^{-\gamma_i(t-s)}\,\mathrm{d}W(s)\int_0^\tau e^{-\gamma_j(\tau-s)}\,\mathrm{d}W(s)\right] \tag{53}$$

$$= \sum_{i,j} \omega_i \omega_j \int_0^\tau e^{-\gamma_i(t-s)-\gamma_j(\tau-s)}ds \tag{54}$$

$$= \sum_{i,j} \omega_i \omega_j \left(\frac{e^{-\gamma_i(t-\tau)}}{\gamma_i+\gamma_j} - \frac{e^{-\gamma_i t-\gamma_j \tau}}{\gamma_i+\gamma_j}\right) \tag{55}$$

**fBM and MA-fBM (Type I).** Since (Dfn. 3)

$$\hat{B}_H^{(I)}(t) = \sum_k \omega_k(Y_k(t) - Y_k(0))$$

where (Eq. (6))

$$Y_k(t) - Y_k(0) = Y_k(0)(e^{-\gamma_k t} - 1) + \int_0^t e^{-\gamma_k(t-s)}\,\mathrm{d}W(s)$$

and (Eq. (25))

$$Y_k(0) = \int_{-\infty}^0 e^{\gamma_k s}\,\mathrm{d}W(s)\,.$$

we can write

$$\hat{B}_H^{(I)}(t) = \sum_k \omega_k\left((e^{-\gamma_k t}-1)\int_{-\infty}^0 e^{\gamma_k s}\,\mathrm{d}W(s) + \int_0^t e^{-\gamma_k(t-s)}\,\mathrm{d}W(s)\right)\,. \tag{56}$$

This leads to the following derivation (using Itô isometry (Øksendal & Øksendal, 2003)):

$$\mathbb{E}\left[\hat{B}_H^{(I)}(t)B_H^{(I)}(t)\right] = \frac{1}{\Gamma(H+1/2)}\sum_k \omega_k \mathbb{E}\Bigg[\left(\int_{-\infty}^0 \left((t-s)^{H-1/2} - (-s)^{H-1/2}\right)\mathrm{d}W(s)\right.$$

$$\left. + \int_0^t (t-s)^{H-1/2}\,\mathrm{d}W(s)\right)$$

$$\cdot \left((e^{-\gamma_k t}-1)\int_{-\infty}^0 e^{\gamma_k s}\,\mathrm{d}W(s) + \int_0^t e^{-\gamma_k(t-s)}\,\mathrm{d}W(s)\right)\Bigg] \tag{57}$$

$$= \frac{1}{\Gamma(H+1/2)}\sum_k \omega_k \Bigg((e^{-\gamma_k t}-1)\int_{-\infty}^0 \left((t-s)^{H-1/2} - (-s)^{H-1/2}\right)e^{\gamma_k s}\,\mathrm{d}s$$

$$+ \int_0^t (t-s)^{H-1/2}e^{-\gamma_k(t-s)}\,\mathrm{d}s\Bigg) \tag{58}$$

$$= \sum_k \omega_k \frac{2 - e^{-\gamma_k t} - Q(H+1/2, \gamma_k t)e^{\gamma_k t}}{\gamma_k^{H+1/2}} \tag{59}$$

where $Q(z,x) = \frac{1}{\Gamma(z)}\int_x^\infty t^{z-1}e^{-t}\,\mathrm{d}t$ is the regularized upper incomplete gamma function.

**fBM and MA-fBM (Type II).**

$$\mathbb{E}\left[\hat{B}_H^{(II)}(t)B_H^{(II)}(t)\right] = \frac{1}{\Gamma(H+1/2)}\sum_k \omega_k \mathbb{E}\left[\int_0^t e^{-\gamma_k(t-s)}\,\mathrm{d}W(s)\int_0^t (t-s)^{H-1/2}\,\mathrm{d}s\right] \quad (60)$$

$$= \frac{1}{\Gamma(H+1/2)}\sum_k \omega_k \int_0^t e^{-\gamma_k(t-s)}(t-s)^{H-1/2}\,\mathrm{d}s \quad (61)$$

$$= \sum_k \omega_k \frac{P(H+1/2,\gamma_k t)}{\gamma_k^{H+1/2}} \quad (62)$$

where $P(z,x) = \frac{1}{\Gamma(z)}\int_0^x t^{z-1}e^{-t}\,\mathrm{d}t$ is the regularized lower incomplete gamma function.

## D  CHOOSING $\omega_k$ VALUES

### D.1  BASELINE

To approximate the integral in equation (8) for $H < 1/2$ we do a piece-wise linear approximation of the integral between the known $Y_k(t)$ values:

$$\sum_{k=1}^K \omega_k Y_k(t) = \sum_{k=1}^{K-1}\int_{\gamma_k}^{\gamma_{k+1}}\left(\frac{\gamma_{k+1}-\gamma}{\gamma_{k+1}-\gamma_k}Y_k(t) + \frac{\gamma-\gamma_k}{\gamma_{k+1}-\gamma_k}Y_{k+1}(t)\right)\mu(\gamma)\,\mathrm{d}\gamma \quad (63)$$

For $H > 1/2$ we approximate $\partial_\gamma Y_\gamma(t)$ with finite differences:

$$\sum_{k=1}^K \omega_k Y_k(t) = \sum_{k=1}^{K-1} -\frac{Y_{k+1}(t)-Y_k(t)}{\gamma_{k+1}-\gamma_k}\int_{\gamma_k}^{\gamma_{k+1}}\nu(\gamma)\,\mathrm{d}\gamma \quad (64)$$

This leads to the following proposal for $\omega_k$:

$$\omega_k = \begin{cases} \dfrac{1}{\Gamma(\alpha)\Gamma(1-\alpha)}\left(\mathbf{1}_{k>1}\dfrac{\frac{\gamma_k^{2-\alpha}-\gamma_{k-1}^{2-\alpha}}{2-\alpha}-\gamma_{k-1}\frac{\gamma_k^{1-\alpha}-\gamma_{k-1}^{1-\alpha}}{1-\alpha}}{\gamma_k-\gamma_{k-1}}\right. \\ \qquad\qquad\qquad \left. +\mathbf{1}_{k<K}\dfrac{\gamma_{k+1}\frac{\gamma_{k+1}^{1-\alpha}-\gamma_k^{1-\alpha}}{1-\alpha}-\frac{\gamma_{k+1}^{2-\alpha}-\gamma_k^{2-\alpha}}{2-\alpha}}{\gamma_{k+1}-\gamma_k}\right), & H < 1/2 \\[2em] \dfrac{-1}{(2-\alpha)\Gamma(\alpha)\Gamma(2-\alpha)}\left(\mathbf{1}_{k>1}\dfrac{\gamma_k^{2-\alpha}-\gamma_{k-1}^{2-\alpha}}{\gamma_k-\gamma_{k-1}}-\mathbf{1}_{k<K}\dfrac{\gamma_{k+1}^{2-\alpha}-\gamma_k^{2-\alpha}}{\gamma_{k+1}-\gamma_k}\right), & H > 1/2 \end{cases} \quad (65)$$

where $\alpha = H + 1/2$.

### D.2  A PROOF FOR THE OPTIMIZED $\omega_k$ VALUES

To optimize $\omega_k$ values, we first provide a closed form expression for the approximation error and then show how we can solve for the $\omega_k$ that minimize this error.

**Type I.** We will start by optimizing $\omega_k$ for Type I. Consider the error:

$$\mathcal{E}^{(I)}(\boldsymbol{\omega}) = \int_0^T \mathbb{E}\left[\left(\hat{B}_H^{(I)}(t) - B_H^{(I)}(t)\right)^2\right]\mathrm{d}t \quad (66)$$

$$= \int_0^T\left(\mathbb{E}\left[\hat{B}_H^{(I)}(t)^2\right] + \mathbb{E}\left[B_H^{(I)}(t)^2\right] - 2\mathbb{E}\left[\hat{B}_H^{(I)}(t)B_H^{(I)}(t)\right]\right)\mathrm{d}t \quad (67)$$

Using Eqs. (1), (50) and (59)

$$
\mathcal{E}^{(I)}(\boldsymbol{\omega}) = \int_0^T \left( \sum_{i,j} \omega_i \omega_j \frac{2 - e^{-\gamma_i t} - e^{-\gamma_j t}}{\gamma_i + \gamma_j} + t^{2H} \right.
$$
$$
\left. - 2 \sum_k \omega_k \frac{2 - e^{-\gamma_k t} - Q\left(H + 1/2, \gamma_k t\right) e^{\gamma_k t}}{\gamma_k^{H+1/2}} \right) \mathrm{d}t \tag{68}
$$
$$
= \sum_{i,j} \omega_i \omega_j \frac{2T + \frac{e^{-\gamma_i T} - 1}{\gamma_i} + \frac{e^{-\gamma_j T} - 1}{\gamma_j}}{\gamma_i + \gamma_j} + \frac{T^{2H+1}}{2H+1}
$$
$$
- 2 \sum_k \omega_k \left( \frac{2T}{\gamma_k^{H+1/2}} - \frac{T^{H+1/2}}{\gamma_k \Gamma(H + 3/2)} + \frac{e^{-\gamma_k T} - Q\left(H + 1/2, \gamma_k T\right) e^{\gamma_k T}}{\gamma_k^{H+3/2}} \right) \tag{69}
$$

This leads to the quadratic form $\mathcal{E}^{(I)}(\boldsymbol{\omega}) = \boldsymbol{\omega}^T \boldsymbol{A}^{(I)} \boldsymbol{\omega} - 2\boldsymbol{b}^{(I)T} \boldsymbol{\omega} + c^{(I)}$ with

$$
\boldsymbol{A}_{i,j}^{(I)} = \frac{2T + \frac{e^{-\gamma_i T} - 1}{\gamma_i} + \frac{e^{-\gamma_j T} - 1}{\gamma_j}}{\gamma_i + \gamma_j} \tag{70}
$$
$$
\boldsymbol{b}_k^{(I)} = \frac{2T}{\gamma_k^{H+1/2}} - \frac{T^{H+1/2}}{\gamma_k \Gamma(H + 3/2)} + \frac{e^{-\gamma_k T} - Q(H + 1/2, \gamma_k T) e^{\gamma_k T}}{\gamma_k^{H+3/2}} \tag{71}
$$
$$
c^{(I)} = \frac{T^{2H+1}}{2H+1}. \tag{72}
$$

**Type II**. We now repeat a similar procedure for the Type II.

$$
\mathcal{E}^{(II)}(\boldsymbol{\omega}) = \int_0^T \mathbb{E}\left[ \left( \hat{B}_H^{(II)}(t) - B_H^{(II)}(t) \right)^2 \right] \mathrm{d}t \tag{73}
$$
$$
= \int_0^T \left( \mathbb{E}\left[ \hat{B}_H^{(II)}(t)^2 \right] + \mathbb{E}\left[ B_H^{(II)}(t)^2 \right] - 2\mathbb{E}\left[ \hat{B}_H^{(II)}(t) B_H^{(II)}(t) \right] \right) \mathrm{d}t \tag{74}
$$

Using Eqs. (2), (55) and (62)

$$
\mathcal{E}^{(II)}(\boldsymbol{\omega}) = \int_0^T \sum_{i,j} \omega_i \omega_j \frac{1 - e^{-(\gamma_i + \gamma_j)t}}{\gamma_i + \gamma_j} + \frac{t^{2H}}{2H\Gamma(H + 1/2)^2} - 2 \sum_k \omega_k \frac{P\left(H + 1/2, \gamma_k t\right)}{\gamma_k^{H+1/2}} \, \mathrm{d}t \tag{75}
$$
$$
= \sum_{i,j} \omega_i \omega_j \frac{T + \frac{e^{-(\gamma_i + \gamma_j)T} - 1}{\gamma_i + \gamma_j}}{\gamma_i + \gamma_j} + \frac{T^{2H+1}}{2H(2H + 1)\Gamma(H + 1/2)^2} \tag{76}
$$
$$
- 2 \sum_k \omega_k \left( \frac{T}{\gamma_k^{H+1/2}} P\left(H + 1/2, \gamma_k T\right) - \frac{H + 1/2}{\gamma_k^{H+3/2}} P\left(H + 3/2, \gamma_k T\right) \right) \tag{77}
$$

This leads to the quadratic form $\mathcal{E}^{(II)}(\boldsymbol{\omega}) = \boldsymbol{\omega}^T \boldsymbol{A}^{(II)} \boldsymbol{\omega} - 2\boldsymbol{b}^{(II)T} \boldsymbol{\omega} + c^{(II)}$ with

$$
\boldsymbol{A}_{i,j}^{(II)} = \frac{T + \frac{e^{-(\gamma_i + \gamma_j)T} - 1}{\gamma_i + \gamma_j}}{\gamma_i + \gamma_j} \tag{78}
$$
$$
\boldsymbol{b}_k^{(II)} = \frac{T}{\gamma_k^{H+1/2}} P(H + 1/2, \gamma_k T) - \frac{H + 1/2}{\gamma_k^{H+3/2}} P(H + 3/2, \gamma_k T) \tag{79}
$$
$$
c^{(II)} = \frac{T^{2H+1}}{2H(2H + 1)\Gamma(H + 1/2)^2}. \tag{80}
$$

**Exactly one solution for $\boldsymbol{\omega}$**. There is exactly one solution if $\boldsymbol{A}^{(I,II)}$ is positive definite, which is defined as

$$
\boldsymbol{\omega}^T \boldsymbol{A}^{(I,II)} \boldsymbol{\omega} > 0 \text{ for all } \boldsymbol{\omega} \in \mathbb{R}^K \setminus \{\boldsymbol{0}\}. \tag{81}
$$

Recall that

$$\boldsymbol{\omega}^T \boldsymbol{A}^{(I,II)} \boldsymbol{\omega} = \int_0^T \mathbb{E}\left[\hat{B}_H^{(I,II)}(t)^2\right] \mathrm{d}t \tag{82}$$

thus there is exactly one solution if

$$\int_0^T \mathbb{E}\left[\hat{B}_H^{(I,II)}(t)^2\right] \mathrm{d}t > 0 \text{ for all } \boldsymbol{\omega} \in \mathbb{R}^K \setminus \{\mathbf{0}\}. \tag{83}$$

Recall that $\hat{B}_H^{(I,II)}(t)$ is a linear combination of $K$ Ornstein-Uhlenbeck processes with speed of mean reversion $\gamma_k$ driven by the same Brownian motion. Under the trivial conditions that $\gamma_i \neq \gamma_j$ (so they can not cancel out) and $\gamma_k < \infty$, this will never be 0. Hence the last inequality holds unless $\hat{B}_H^{(I,II)}(t) = 0$. This concludes the proof.

### D.3 NUMERICALLY STABLE IMPLEMENTATION OF $Q(z,x)e^x$

The term $Q(H + 1/2, \gamma_k T)e^{\gamma_k T}$ in Prop. 5 leads to numerical instability, since $\gamma_k T$ is typically a high number (for the highest $\gamma_k$). On the other hand, $Q(H + 1/2, \gamma_k T)$ is a low number for high $\gamma_k T$. Our stable implementation makes use of a continued fraction (Cuyt et al., 2008, eq. (12.6.17)), using the 'Kettenbruch' notation (Cuyt et al., 2008, sec. 1.1) for continued fractions:

$$Q(H + 1/2, \gamma_k T)e^{\gamma_k T} = \frac{\Gamma(H + 1/2, \gamma_k T)}{\Gamma(H + 1/2)} e^{\gamma_k T} \tag{84}$$

$$= \frac{1}{\Gamma(H + 1/2)(\gamma_k T)^{H+1/2}} \overset{\infty}{\underset{m=1}{\mathrm{K}}} \left(\frac{a_m(H + 1/2)/(\gamma_k T)}{1}\right) \tag{85}$$

where $a_m(a)$ is given by

$$a_1(a) = 1, \quad a_{2j}(a) = j - a, \quad a_{2j+1}(a) = j, \quad j \geq 1 \tag{86}$$

In practice we observe better accuracy with the original equation for $\gamma_k T < 10$, where it is still stable, and only need 5 fractions to approximate the equation for $\gamma_k T > 10$.

## E DETAILS ON MODEL ARCHITECTURES & HYPERPARAMETERS

### E.1 FOU BRIDGE

For all experiments, $K = 5$ and $\gamma_k = (\frac{1}{20}, \ldots, 20)$. We use "Type I" and the optimal definitions for $\omega_k$, with a time horizon $T = 6$. The control function is a neural network with two hidden layers of each 1000 neurons, with $\tanh$ activation function. Its input is represented as $[\sin t, \cos t, X(t), Y_1(t), \ldots, Y_K(t)]$. The control function is initialized so that its output is 0 at the start of training. Models are trained for 2000 training steps with a batch size of 32. We use the Adam (Kingma & Ba, 2014) optimizer with fixed learning rate $10^{-3}$. We use the *Stratonovich–Milstein* SDE solver (Kidger, 2021) with an integration step of 0.01. The length of the bridge $T = 2$ and observation noise $\sigma = 0.1$.

### E.2 TIME DEPENDENT HURST INDEX

We directly compare our method with the data and estimate found in the published codebase of Tong et al. (2022)[3]. We choose $K = 5$ and $\gamma_k = (\frac{1}{20}, \ldots, 20)$ and use "Type II" (to match the data and noise type in Tong et al. (2022)). The optimal definitions for $\omega_k$, with time horizon $T = 2$ are used. The control function is a neural network with two hidden layers of each 1000 neurons, with $\tanh$ activation function. Its input is represented as $[\sin t, \cos t, \sin 2t, \cos 2t, \ldots, \sin 5t, \cos 5t, X(t), Y_1(t), \ldots, Y_K(t)]$. The model is trained for 1000 training steps with a batch size of 4. We use the Adam (Kingma & Ba, 2014) optimizer with a learning rate $3 \times 10^{-3}$, scheduled with cosine decay to $3 \times 10^{-4}$ by the end of training. We use the *Stratonovich-Milstein* SDE solver (Kidger, 2021). The integration step is 0.005 and observation noise $\sigma = 0.025$ (both identical to Tong et al. (2022)).

---

[3] https://github.com/anh-tong/fractional_neural_sde/blob/7565a2/ fractional_neural_sde/example.ipynb

### E.3 LATENT VIDEO MODEL

**Stochastic moving MNIST**. For the MA-fBM model, $K = 5$ and $\gamma_k = (\frac{1}{20}, \ldots, 20)$. We use "Type I" and the corresponding definitions for $\omega_k$, with a time horizon $T = 2.4$. For the BM model, $K = 1$, $\gamma_1 = 0$ and $\omega = 1$, which naturally corresponds to white Brownian motion. The number of latent dimensions $D = 6$.

The encoder model consists of four blocks, containing a convolution layer, maxpool, groupnorm and SiLU activation. Each block reduces spatial dimension by 2, and the number of features in each block is $(64, 128, 256, 256)$. The last output is flattened and is the input of a dense layer, with $h$ as output with 64 features.

The median over the time axis of $h$ is fed into a two layers neural network to produce the static content vector $w$. Since the median is permutation invariant, $w$ contains no dynamic information, only static information. $w$ also has 64 features.

The context model consists of two subsequent $1 - D$ convolutions in the temporal dimension. Thus, information is shared over different frames, which is necessary for inference. The output of this model is $g$.

To start the SDE integration, we need an initial state that is conditioned on the data. We define a three layer neural network model that receives $(g_1, h_1, h_2, h_3)$ and outputs the parameters of the posterior distribution $q_{x_1}$ of the initial state of the SDE. $x_1$ is sampled from $q_{x_1}$, which we model as a diagonal Normal distribution. The parameters of a prior model $p_{x_1}$ are also optimized, and the Kullback-Leibler divergence $D_{\mathrm{KL}}(p_{x_1}, q_{x_1})$ is added to the loss function. This approach for training neural SDEs is similar to others in literature (Li et al., 2020).

The prior drift $b_\theta(X, t)$ and the control function $u(Z(t), t)$ have the same architecture, a neural network with two hidden layers of each 200 neurons, with $\tanh$ activation functions. The shared diffusion $\sigma_\theta(X, t)$ is implemented so that the noise is commutative to allow Milstein solvers (Li et al., 2020; Kidger et al., 2021), *i.e.* $\sigma_\theta(X, t)$ is diagonal and the $i$-th component on the diagonal only receives $X_i(t)$ as input, where we have defined $D$ separate neural networks for each component. Each neural network has two layers with 200 neurons and $\tanh$ activations.

$b_\theta$ and $\sigma_\theta$ receive $X(t)$ as input. The control function a concatenated vector of $(X(t), Y_1(t), \ldots, Y_K(t), g(t))$. $g(t)$ is a linear interpolation of $g$ at time $t$. This enables the control function to use appropriate information to be able to steer the process correctly.

The resulting states $x$ after integration of the SDE are fed, together with the static content vector $w$ in the decoder model. The decoder model has first a dense layer. The outputs of this first layer are shaped in a $4 \times 4$ spatial grad. Subsequently, four blocks with a convolution layer, groupnorm, a spatial nearest neighbour upsampling layer and a SiLU activation. Thus, the model reaches the correct resolution of $64 \times 64$. Two additional convolution layers with SiLU activation and a final sigmoid activation complete the decoder model.

We train on sequences of 25 frames, with a time length of 2.4 (0.1 per frame). The frames have resolution $64 \times 64$ and 1 color channel. Each model was trained for 187500 training steps with a batch size of 32. We use the Adam (Kingma & Ba, 2014) optimizer with fixed learning rate $3 \times 10^{-4}$. We use the *Stratonovich–Milstein* SDE solver (Kidger, 2021) with an integration step of 0.033 (3 integration steps per data frame). Models were trained on a single NVIDIA GeForce RTX 4090, which takes around 39 hours for one model.

**Double pendulum**. We use the train-test split from the original dataset (Asseman et al., 2018). The videos are recorded with a high speed camera, we used every 10th frame to increase the challenge of the dataset. We resized the frames to a resolution of $128 \times 128$ resolution. Therefore, we added one block to the encoder and decoder model to achieve this resolution, compared to the model for stochastic moving MNIST. We did not use the static content vector $w$, since there is minimal static information in this dataset, and used $D = 8$ latent dimensions. The models were trained for 124916 training steps, and we trained around 32 hours for one model. Beyond these outlined differences, all other details are equal to the stochastic moving MNIST model.

# F  ADDITIONAL EXPERIMENTAL RESULTS

## F.1  GENERATED TRAJECTORIES OF MA-fBM FOR VARYING $K$

Included here are some of the trajectories used to calculate the MSE of the generated trajectories for MA-fBM for varying $K$ (Fig. 7). We show trajectories of MA-fBM with our approach (Sec. 4.1) and the baseline method (*cf.* App. D.1) for choosing $\omega_k$. True paths are plotted in black, the approximations with varying $K$ in a color-scale as indicated in the legends, see Figs. 9 to 12 and 14 to 16. Our method quickly converges to the true path for increasing $K$, while much slower for the baseline method.

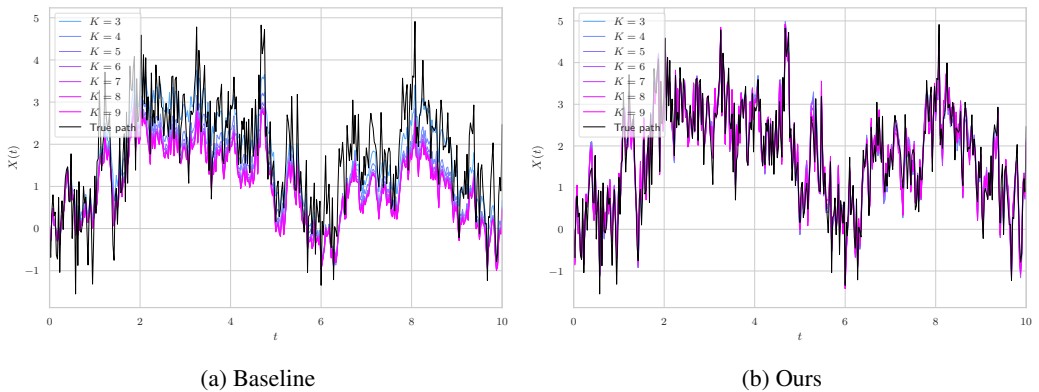

(a) Baseline                  (b) Ours

Figure 9: Generated trajectories of (**a**) the baseline method summarized in App. D.1 and (**b**) our method, MA-fBM (Sec. 4.1), for varying $K$ and $H = 0.1$.

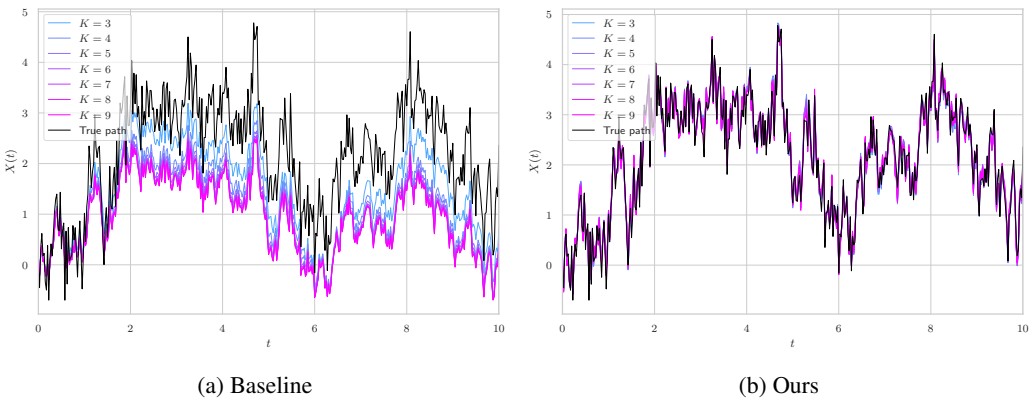

(a) Baseline                  (b) Ours

Figure 10: Generated trajectories of (**a**) the baseline method summarized in App. D.1 and (**b**) our method, MA-fBM (Sec. 4.1), for varying $K$ and $H = 0.2$.

## F.2  fOU BRIDGE

Fig. 17 shows additional results of the fractional Ornstein–Uhlenbeck bridge. The variances are calculated with Eq. (23), and Eq. (24) for $\theta > 0$ and $H > 1/2$ or Eq. (1) for $\theta = 0$. Note that we do not have a useful covariance equation for $\theta > 0$ and $H < 1/2$ (Lysy & Pillai, 2013), so this setting is not included in the experiments.

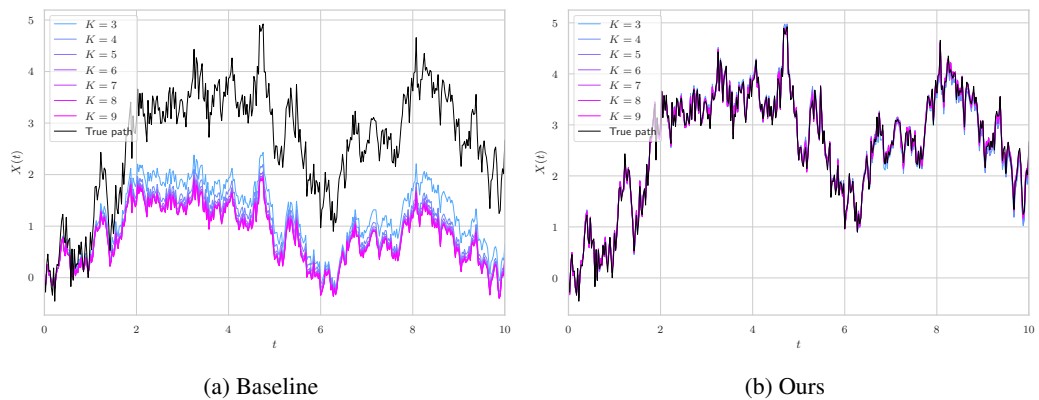

(a) Baseline  (b) Ours

Figure 11: Generated trajectories of (**a**) the baseline method summarized in App. D.1 and (**b**) our method, MA-fBM (Sec. 4.1), for varying $K$ and $H = 0.3$.

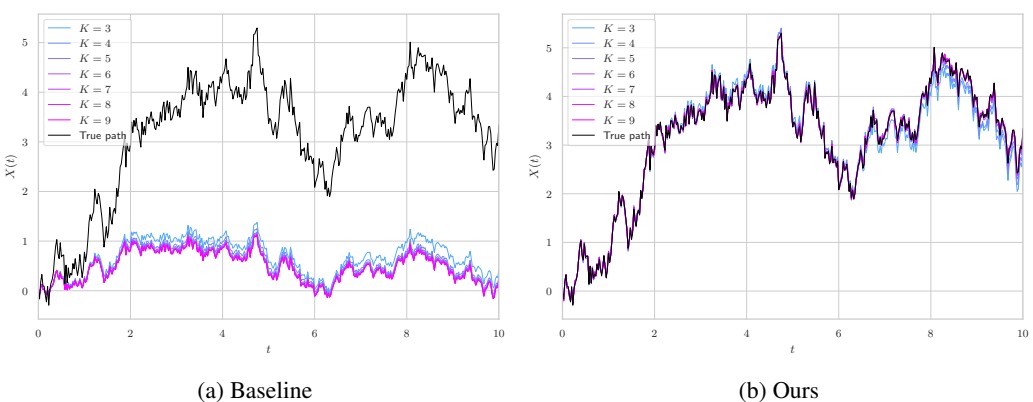

(a) Baseline  (b) Ours

Figure 12: Generated trajectories of (**a**) the baseline method summarized in App. D.1 and (**b**) our method, MA-fBM (Sec. 4.1), for varying $K$ and $H = 0.4$.

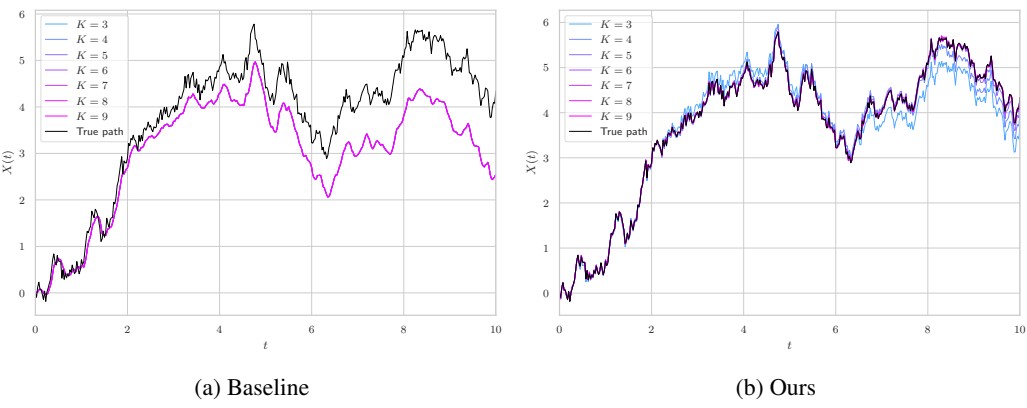

(a) Baseline  (b) Ours

Figure 13: Generated trajectories of (**a**) the baseline method summarized in App. D.1 and (**b**) our method, MA-fBM (Sec. 4.1), for varying $K$ and $H = 0.5$.

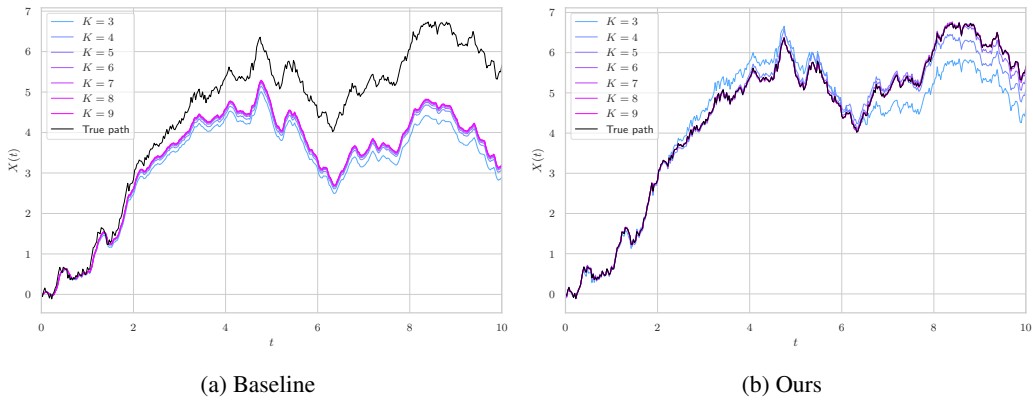

(a) Baseline        (b) Ours

Figure 14: Generated trajectories of (**a**) the baseline method summarized in App. D.1 and (**b**) our method, MA-fBM (Sec. 4.1), for varying $K$ and $H = 0.6$.

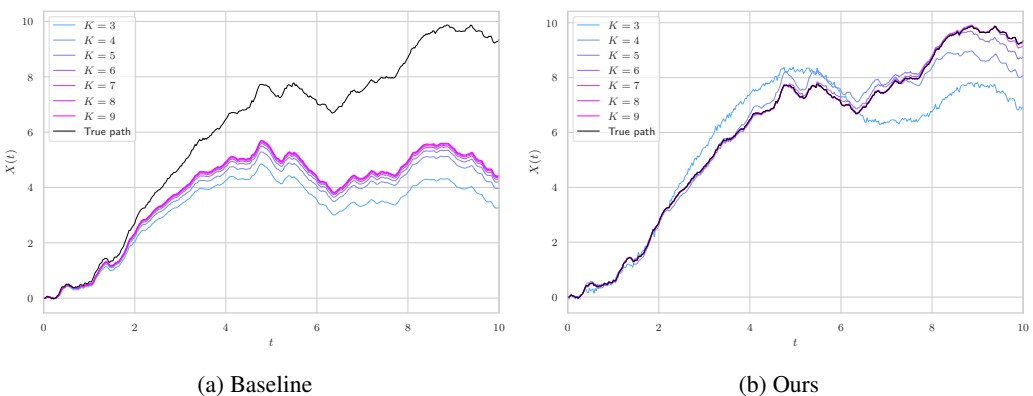

(a) Baseline        (b) Ours

Figure 15: Generated trajectories of (**a**) the baseline method summarized in App. D.1 and (**b**) our method, MA-fBM (Sec. 4.1), for varying $K$ and $H = 0.8$.

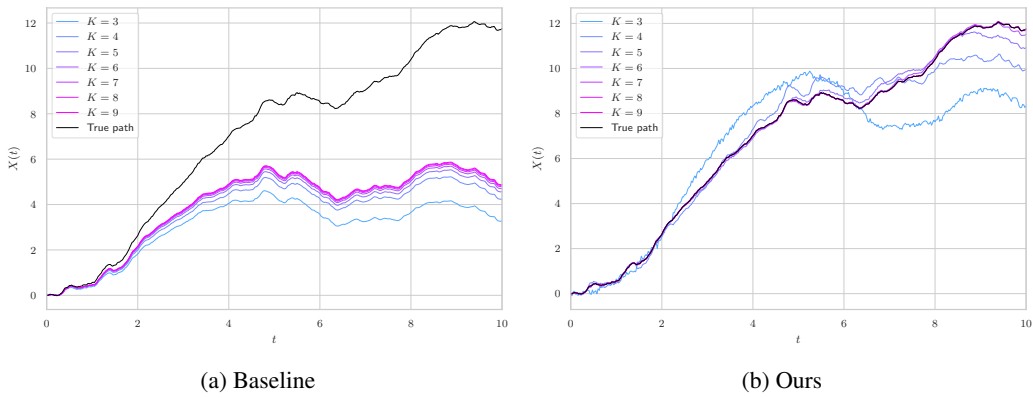

(a) Baseline        (b) Ours

Figure 16: Generated trajectories of (**a**) the baseline method summarized in App. D.1 and (**b**) our method, MA-fBM (Sec. 4.1), for varying $K$ and $H = 0.9$.

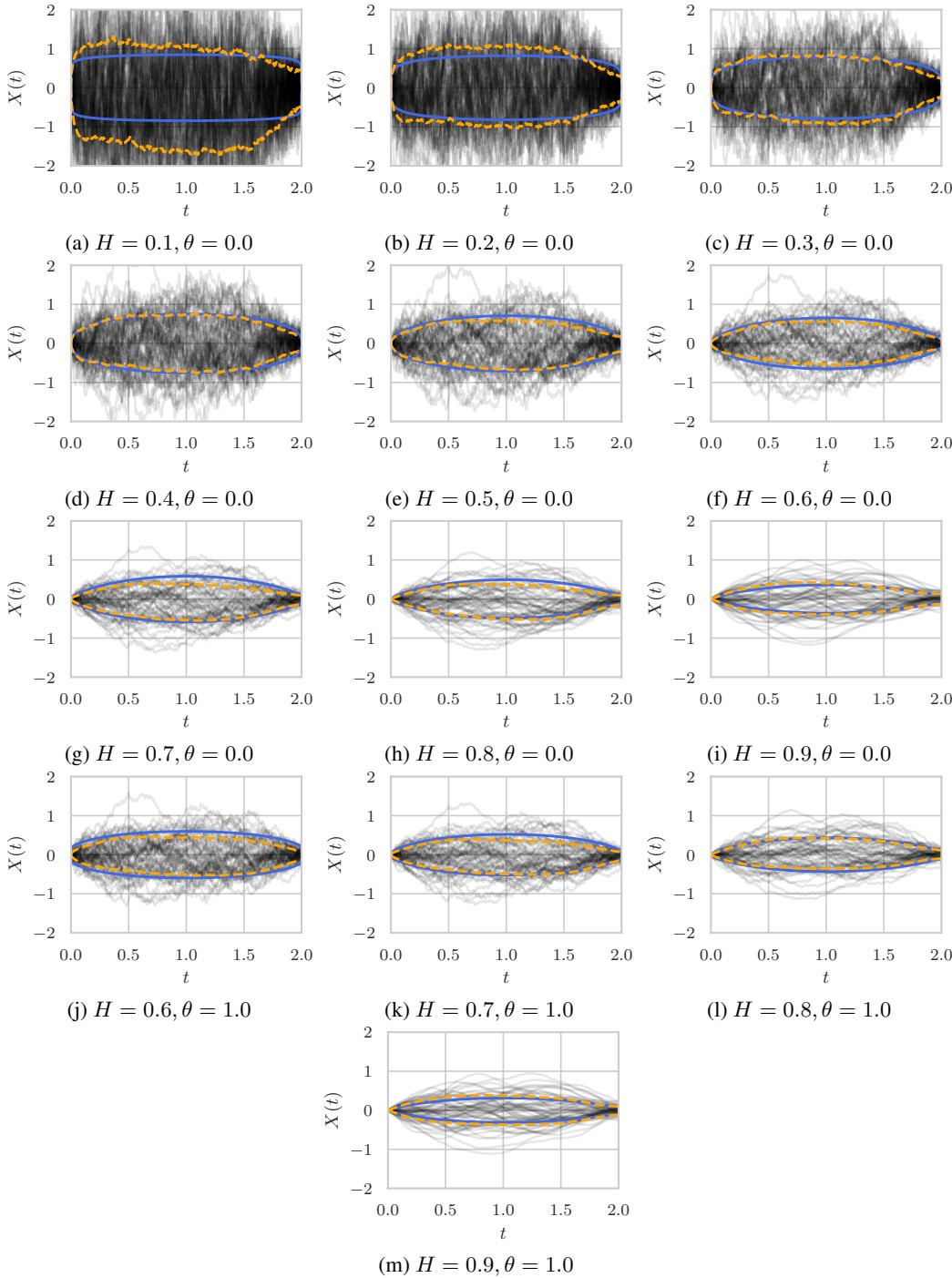

Figure 17: The true variance (blue) of a fOU bridge matches the empirical variance (dashed orange) of our trained models. The transparent black lines are the sampled approximate posterior paths used to calculate the empirical variance.

### F.3 Video Models

**On the choice of video datasets**. We conducted experiments on two video datasets: Stochastic Moving MNIST (SM-MNIST) (Denton & Fergus, 2018) and the real video dataset of a chaotic double pendulum (Asseman et al., 2018).

SM-MNIST and the double pendulum dataset contain different forms of nuisances and present different challenges to our stochastic model. First, SM-MNIST digits move with a constant velocity along a trajectory until they hit at wall at which point they bounce off with a random speed and direction. This sudden event intersperses the deterministic motion with moments of uncertainty, i.e. each time a digit hits a wall. This is the reason why a stochastic model fits better than an ODE and unlike BM, our noise can model the smooth and correlated trajectory simply by raising the Hurst index.

On the other hand, the double pendulum dataset is actually governed by a set of coupled ordinary differential equations. However, despite being a simple physical system, it exhibits a rich dynamic behavior with a strong sensitivity to initial conditions and noises in the environment (motion of the air in the room, sound vibrations, vibration of the table due to coupling with the pendulum etc.). Combined with the chaotic nature of the system, this creates a major challenge for any model based upon smooth ODEs. Our model on the other hand heavy lifts this difficulty onto the (fractional) stochastic noise, leading to a more appropriate model. As shown in Tab. 2, our model outperforms the BM baseline also in this dataset.

Figs. 18 and 19 show the posterior reconstructions of models trained on the Stochastic Moving MNIST and the double pendulum dataset respectively. Fig. 20 shows stochastic video prediction samples of Stochastic Moving MNIST.

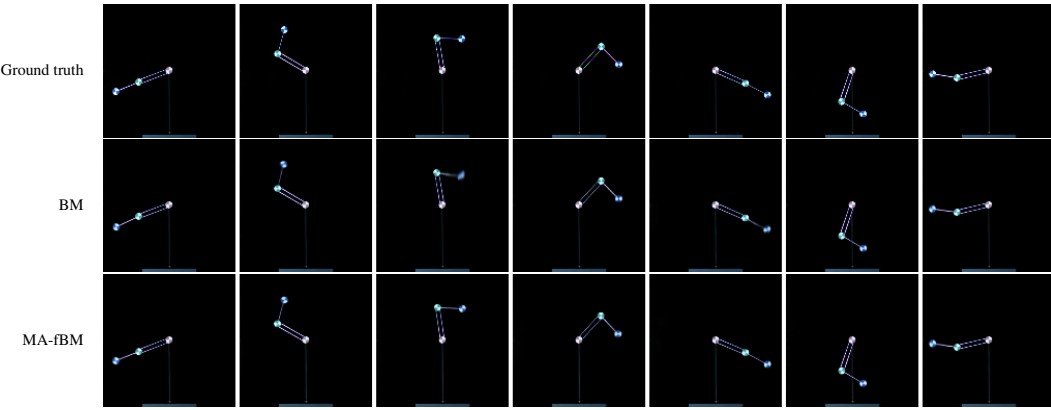

Figure 18: Posterior reconstructions of a model driven by BM and a model driven by MA-fBM, conditioned on the same data ('Ground truth').

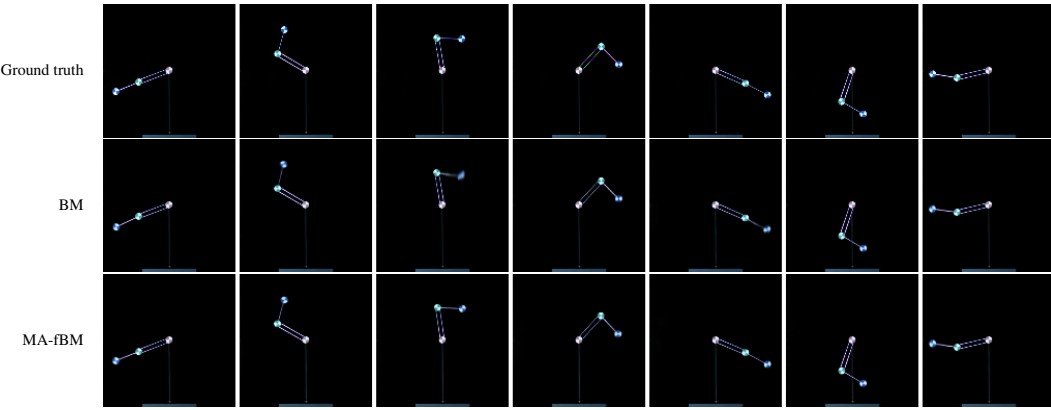

Figure 19: Posterior reconstructions of a model driven by BM and a model driven by MA-fBM, trained on the double pendulum dataset. Both are conditioned on the same data ('Ground truth'). We show 7 evenly spaced frames of the total 20 frames.

Figure 20: Stochastic predictions using the trained prior of a model driven by BM and a model driven by MA-fBM, where the initial state is conditioned on the same data. Four samples are shown for each model. The MA-fBM samples show more diverse movements, thus better capturing the dynamics in the data. The BM samples are more similar, indicating a less powerful prior was learned.

