# OpenReview forum: "Variational Inference for SDEs Driven by Fractional Noise"
_ICLR.cc/2024/Conference — ICLR 2024 spotlight_

### Official Review · Reviewer_kkY2 · 2023-10-25

**Soundness:** 3 good
**Presentation:** 4 excellent
**Contribution:** 3 good
**Rating:** 8
**Confidence:** 5

**Summary:**

The paper presents a novel variational inference method for stochastic differential equations (SDEs) under fractional Brownian motions (fBMs). The key property of fBMs lies in long-term dependence in its stochastic generation. In order to devise the proposed variational inference, the paper suggests to use approximation based on the result of Harms & Stefanovits. That is, fBMs can be represented as integral (infinite linear combination) via Ornstein-Uhlenbeck processes. The SDEs under fBMs are then rewritten as a system of SDEs under Brownian noise.

**Strengths:**

- The paper is overall well-written, providing an excellent introduction of fractional Brownian motion including type I, and type II.
- Showcase a practical use of Harms & Stefanovits, 2019. This is a nice approach compared to Tong et al. 2022 suffered from some limitations including the choice of solvers.
- Interesting application learning for SDE.

**Weaknesses:**

- Perhaps variational inference may not be the main contribution of the paper since it is still a result of Girsanov's theorem like presented in Li et al. 2020. However, I believe that the idea of finding SDE approximation outweights this point.

**Questions:**

Minor points:
-  In the first paragraph of Background section, do you mean "learning comunity" as "machine learning comunity"?
- At Proposition 1, “Markov rocesses" should be “Markov processes”.
- At Proposition 2, "R^D" should be "\mathbb{R}^D".

---

> ### Author Response · Authors · 2023-11-22
> **Author Response**
>
> We thank the reviewer for their helpful comments and appreciate that they find our introduction to be excellent. We agree with the reviewer that our variational inference resembles earlier works such as Li et al. (2020) and is a direct application of Girsanov's theorem. However, to benefit from this strong, established result, we did need to show how a new SDE driven by BM can be obtained from an SDE driven by fBM. As the reviewer confirms, this is where the novelty lies (in addition to our novel scheme of weight optimization). Moreover, our variational inference framework paves the way to our video prediction application. We have incorporated all the minor suggestions.

---

### Official Review · Reviewer_9F2a · 2023-10-30

**Soundness:** 3 good
**Presentation:** 3 good
**Contribution:** 3 good
**Rating:** 8
**Confidence:** 3

**Summary:**

Generative modeling of sequential data (i.e., time series data) is a vibrant research area and has made remarkable progress since the introduction of neural SDEs in recent years. Related to this are variational Bayesian inference methods combined with the application of Girsanov’s change of measure theorem, which have paved the way for learning representative function distributions through stochastic gradient descent. Most of the progress in this area has been on models based on the use of (standard) Brownian motion, i.e., a Markov process. The present work extends existing approaches by building on *fractional* Brownian motion (fMB; a non-Markovian process), questioning the everlasting validity of the Markov property assumption. The key to success lies in the application of Markovian embedding of the fBM and its strong approximation which makes the traditional machinery of neural SDEs accessible for working with non-Markovian systems. Conducted experiments on real data sets as well as ablation studies provide empirical evidence for doubts about the unrestricted model suitability of previous approaches in real world applications.

**Strengths:**

I enjoyed reading this paper because (i) I like the idea of challenging common theoretical assumptions (in this case, the Markov property), (ii) the very well-organized writing style that makes it easy to follow the arguments and (iii) the thorough presentation of the theoretical aspects. Further, the paper content is original to the best of my knowledge and well positioned to existing work, as well as with very few exceptions, has no spelling or grammatical flaws. Empirical evidence is accessed through experiments on synthetical and real-world video data. The results are on a par with compared existing methods and reproducibility is granted by submitted code files. Finally, on a positive note, the conjecture that data contain information that cannot be represented by Markov processes is empirically tested by estimating the Hurst index.

**Weaknesses:**

A major drawback of (traditional) NSDE approaches in the context of variational Bayesian inference is the computational and storage costs associated with learning almost arbitrarily complex representative function distributions. Recent approaches address this problem and attempt to find solutions, e.g., (Li et al. 2020) and (Kidger et al. 2021). However, learning large datasets still seems to be very resource intensive. In the context of the present work, I wonder to what extent the additional Markov approximation scheme exacerbates the aforementioned problem. For example, in the case of the video experiment presented, the authors report (p. 19, Appendix E.3) "Models were trained on a single NVIDA GeFORCE RTX 4090, which takes about 39 hours for one model." Can you please provide a more detailed evaluation of runtime in addition, including a comparison to baseline approaches?

**Questions:**

1. Can you elaborate on the difference between a Type I and a Type II fBM and which form delivers benefits in which circumstances?
2. It would be of importance to comment on the assumptions that guarantee a finite KL term inside of the ELBO, i.e., the diffusion terms of prior and posterior have to be equal.
3. According to (Oksendal 2003, The Girsanov Theorem II), the control function $u$ needs to meet Novikov's condition. In the proposed work $u$ is parameterized by a neural net. How is the aforementioned condition controlled during training?
4. Can you elaborate more on the following: "Unlike Tong et al. (2022) our approach is agnostic to discretization and the choice of the solver."

Minor:
- Missing parenthesis in Eq. (5)
- Proposition 1 : Markov rocesses -> Markov processes
- $\text{d} t$ vs $dt$

**Details Of Ethics Concerns:**

--

---

> ### Author Response · Authors · 2023-11-22
> **Author Response 1**
>
> We thank the reviewer for the nice summary of our work as well as their constructive feedback. We appreciate that the reviewer has a good grasp of the essence of our contributions. We address their key comments below, while we have incorporated all of their minor corrections.
>
> $\textbf{On runtime of large video models.}$ As the reviewer rightously confirms, the computational cost is not a drawback specific to our method. Almost all similar pipelines require moderate GPU resources and training time. For example, on UCF101 dataset, a version of TGAN [1] took three days to train on an A100 GPU costing over $350 using Google Cloud Platform [2].
> For KITTI dataset, Akan et al.[3] train on a Tesla V100 GPU for approximately 4-5 days. Similarly they trained SVG and SRVP on a Tesla T4 GPU taking approximately 2 days. Both of these GPUs are almost twice as fast as our RTX 4090.
> Please find a runtime analysis of our method against that of Tong et al. (2022) in our second response to Reviewer J9Kf.
>
> $\textbf{Difference between a Type I and a Type II fBM.}$
> Type I, also called Mandelbrot-Van Ness or 'standard' fBM is the most well known definition of fBM.
> Type II, also called Riemann-Liouville fBM, is historically most prevalent in econometric literature.
> As can be seen in the integral definitions (Eq. (3) and (4)), Type II omits the first integral from $-\infty$ to $0$ in the definition of Type I.
> So Type II is, in a sense, a simplification. Yet, to the best of our knowledge, its covariance has no simple analytical expression. On the other hand, the Type I covariance is the straightforward, well known Eq. (1).
>
> As described by [5,6] the main difference is that Type I has stationary increments, and Type II has non-stationary increments. This means that Type II has a larger emphasis on the origin $t=0$, which might not be favourable for some applications. For example, if during training we sample a sequence from a video dataset at a random start point, this $t=0$ has no special or distinguished meaning and should not be treated differently by the driving fBM process. In other words, Type I ensures a shift in time has no effect on its increments. However, this is not the case for Type II.
> This difference is relevant for our framework, since the increments are driving the SDE.
> We thank the reviewer for bringing this up and have added our explanations to the appendix.
>
> $\textbf{A finite KL term inside of the ELBO and Novikov's conditions.}$
> It is seen in Equations (13) and (16) that the diffusion terms of prior and posterior have to be equal. This holds by construction. However, to derive an explicit finite KL term in our ELBO, we are required to invoke the Girsanov theorem, which necessitates a second technical condition, namely the Novikov condition.
> The reviewer is right that the control term $u(\cdot)$ learned via a neural network is assumed to satisfy Novikov's condition.
> In fact Chen et al. [7] combined with the appendix in https://openreview.net/attachment?id=wG12xUSqrI&name=supplementary_material (Lemma 8) indicates two sufficient conditions for Novikov to hold:
> * sub-Gaussian tails for the initial samples ($X_0$ or $Z_0$), which is easy to enforce
> * uniform Lipschitz continuity of the neural network
>
> The second is challenging for us to guarantee. This is similar to earlier works performing stochastic variational inference, which either use an approximation argument [7] or assumes that Novikov’s condition holds at the outset [8,9,10]. In our work, we admit the latter.
>
> $\textbf{References}$
>
> [1] Saito, M. et al. "Temporal generative adversarial nets with singular value clipping." Proceedings of the IEEE international conference on computer vision. 2017.
>
> [2] Gordon, C., and Parde, N. "Latent neural differential equations for video generation." NeurIPS 2020 Workshop on Pre-registration in Machine Learning. 2021.
>
> [3]  Akan, AK et al. "Stochastic video prediction with structure and motion." arXiv preprint arXiv:2203.10528, 2022.
>
> [5] Lim, SC and Sithi, VM. Asymptotic properties of the fractional brownian motion of riemann-liouville type. Physics Letters A, 1995.
>
> [6] Marinucci M. and Robinson, PM. Alternative forms of fractional brownian motion. Journal of statistical planning and inference, 1999.
>
> [7] Chen, Sitan, et al. "Sampling is as easy as learning the score: theory for diffusion models with minimal data assumptions." The Eleventh International Conference on Learning Representations. 2022.
>
> [8] Li, Xuechen, et al. "Scalable gradients and variational inference for stochastic differential equations." Symposium on Advances in Approximate Bayesian Inference. PMLR, 2020.
>
> [9] Wang, Benjie, et al. "Neural Structure Learning with Stochastic Differential Equations." arXiv preprint arXiv:2311.03309 (2023).
>
> [10] Liu, X. et al. "Let us Build Bridges: Understanding and Extending Diffusion Generative Models." NeurIPS 2022 Workshop on Score-Based Methods. 2022.

---

> ### Author Response · Authors · 2023-11-22
> **Author Response 2**
>
> $\textbf{On discretization and the choice of the solver.}$
> We would like to clarify the distinction to Tong et al. (2022), which mainly differs from our work in two ways: (i) fractional Brownian motion (fBM) is approximated as a a Gaussian process (GP), (ii) only the Type II representation of fBM is used as as an integral over increments of the Wiener process. Tong et al. performs a finite time discretization of the Type II integral to obtain a first approximation of the increments of FBM. In a second step, this approximate GP is further approximated using a sparse GP approach based on a smaller set of pseudo or inducing points which are distributed over time. Conditioned on the inducing points, samples from the sparse GP are independent random variables at each discrete time point. Finally, this (conditioned) white noise process is further interpreted in terms of the Euler discretisation of an ordinary SDE leading to effective drift and diffusions. For the latter SDE, one can apply Girsanov's theorem and the corresponding ELBO (conditioned on the inducing points) to perform inference.
>
> Note, that their current derivation of effective drift and diffusion relies on the Euler discretisation of SDE. For higher order SDE solvers, the approximation has to be adapted, which requires new derivations. As a main difference, in our paper, the approximation is not based on the discretisation in the time domain but of the discretisation of an integral representation (our equation (5)) over a spectrum of decay constants of Ornstein-Uhlenbeck (OU) processes (driven by the same Wiener noise). Since each OU process already represents a noise process with temporal correlations, we can expect that a linear combination of a small number of such processes can yield a good approximation of the covariance of fBM over some given time interval. Our approximation leads to a system of SDEs (without conditioning) for which the ELBO can be easily obtained. Since the time discretisation of the resulting SDE is performed $\textit{after}$ the OU approximation, any SDE solver can be directly applied.
> With this flexibility, in our paper, we have chosen the second order Stratonovich–Milstein solver.

---

> > ### Comment · Reviewer_9F2a · 2023-11-22
> >
> > I thank the authors very much for their thorough response and for the detailed discussion
> > of my questions and concerns.
> >
> > You have clarified things, and I will increase my score correspondingly, under the assumption that these clarifications will make it into the updated version of the paper.
> >
> > Thank you again.

---

> > > ### Author Response · Authors · 2023-11-22
> > >
> > > We thank the reviewer for considering our feedback and positively adapting their evaluation. We kindly note that we have already revised the manuscript and will continue to do so until the final publication.

---

### Official Review · Reviewer_3XSn · 2023-10-31

**Soundness:** 3 good
**Presentation:** 3 good
**Contribution:** 2 fair
**Rating:** 5
**Confidence:** 3

**Summary:**

The paper introduces a novel variational framework for conducting inference in stochastic differential equations driven by Markov-approximate fractional Brownian motion. This framework combines SDEs with variational methods to enable the learning of representative function distributions through stochastic gradient descent. Unlike conventional SDEs that assume Brownian motion, this framework extends to fractional Brownian motion to capture long-term dependencies. Authors derive the evidence lower bound for efficient variational inference and provide a closed-form expression for determining optimal approximation coefficients. The framework is validated on synthetic data, and it also presents a novel architecture for variational latent video prediction.

**Strengths:**

1. This paper introduces generative models that utilize fractional Brownian motion (fBM) as a noise injection, as opposed to conventional score-generative models based on Brownian motion.

2. This paper offers a solution to the limitations of standard Brownian motion by extending the framework to fractional Brownian motion, which better captures long-term dependencies and complexities in real-world data.

3. This paper provides a clear and detailed explanation of the proposed framework and its mathematical foundations.

**Weaknesses:**

1. The experiment lacked sufficient data, and the basis for asserting the presence of long-term dependency was inadequate.

2. An approximation was applied in deriving the theory, and there is no analysis regarding the errors introduced by this approximation.

**Questions:**

1. Why is it believed that Brownian motion cannot capture long-term dependencies, while fBM can?

2. Why does fBM encompass non-Markovian dynamics?

3. Is it necessary to perform the Markov approximation for fBM? What is the extent of error introduced by this, and does it significantly impact accuracy? Did the use of Markov representation not compromise the ability to capture long-term dependencies?

4. Why was latent video generation chosen for experimentation, and why were experiments not conducted on datasets other than MNIST?

---

> ### Author Response · Authors · 2023-11-22
> **Author Response 1**
>
> We thank the reviewer for their constructive feedback and are glad to hear that they find our exposition and mathematical formulation clear and detailed. We respond to their questions and comments below.
>
> $\textbf{On experimentation.}$
> We thank the reviewer for bringing this up. We have now conducted an additional evaluation on the $\textbf{real video dataset of chaotic double pendulum}$, as suggested by the Reviewer J9Kf. Note that this dataset comprises of real video captures of a double pendulum in motion. We provide the details in our first response to J9Kf as well as in our $\textbf{revised Sec. 5 and Appendix}$. Overall, results on this dataset further reveal the merits of fractional noise in capturing the nuisances of real world.
>
> Note that this new dataset and SM-MNIST contain different forms of nuisances and present different challenges to our stochastic model. First, SM-MNIST digits move with a constant velocity along a trajectory until they hit at wall at which point they bounce off with a random speed and direction. This sudden event intersperses the deterministic motion with moments of uncertainty, i.e. each time a digit hits a wall. This is the reason why a stochastic model fits better than an ODE and unlike BM, our noise can model the smooth and correlated trajectory simply by raising the Hurst index.
>
> On the other hand, the double-pendulum dataset is actually governed by a set of coupled ordinary differential equations. However, despite being a simple physical system, it exhibits a rich dynamic behavior with a strong sensitivity to initial conditions and noises in the environment (motion of the air in the room, sound vibrations, vibration of the table due to coupling with the pendulum etc.). Combined with the chaotic nature of the system, this creates a major challenge for any model based upon smooth ODEs. Our model on the other hand heavy lifts this difficulty onto the (fractional) stochastic noise, leading to a more appropriate model. As shown (see our new Table in Sec. 5), our model outperforms the BM baseline also in this dataset.
>
> $\textbf{Why latent video generation?}$
> The main contribution of our work is a new differential equation, driven by the approximate fractional Brownian motion as well as a variational inference framework, whose drift, diffusion and control functions can be learned.
> Due to the stochastic nature of the real world, video perception becomes an appealing and important problem for showcasing the benefits of our method.
> Note that, we do not claim that our video architecture, despite being novel, brings an ultimate solution. Yet, it enables variational learning under our neural SDE, which is essential to validate our contributions. In principle, our stochastic differential equations can be applicable to various sorts of stochastic video perception methods.
> We thank the reviewer for mentioning this and leave further applications to a future study.
>
> $\textbf{BM vs fBM, Markovian-ness and long range dependencies.}$
> Brownian motion is a stochastic process with stationary, independent and normally-distributed increments. Due to this independence, the probability of each event depends only on the state attained and therefore correlations to further away past or future states are not considered, \ie it is a Markov (memory-less) process, $\textit{what happens next depends only on the state of affairs now}$. This can be understood as the noise process (infintesimal increments) being $\delta$-correlated for BM.
> The increments of fBm (with Hurst index $H$), on the other hand, are not independent unless $H=1/2$ for which we recover BM. If $H<1/2$ then the increments of the process are negatively correlated. If $H>1/2$ then the increments of the process are positively correlated, shows $\textit{long-range dependence}$ (LRD), usually related to the $\textit{rate of decay in statistical dependence}$ of two points with increasing time interval. For fBM, the rate of decay follows a power law, indicating that the process is non-Markovian. Please see [1] for a simple yet rigorous proof of this.
>
> In general, we agree that one may have long time dependencies for ordinary stochastic differential equations (without fractional noise). However, for larger classes of such models, e.g. the so called reversible diffusions, one obtains an exponentially fast relaxation of fluctuations under certain asymptotic conditions on potentials (well explained in the book by Pavliotis [2]).
> On the other hand, even simple linear systems driven by fBM, such as the the fOU process, inherit a non exponential, power law decay of stationary correlation functions from the driving process.
>
> $\textbf{References}$
>
> [1] Huy, Dang Phuoc. "A remark on non-Markov property of a fractional Brownian motion." Vietnam Journal of Mathematics 31.3 (2003).
>
> [2] "Stochastic processes and Applications", by G.A. Pavliotis, Springer, Chapter 4.

---

> > ### Author Response · Authors · 2023-11-22
> > **Author Response 2 (Markov Approximation)**
> >
> > $\textbf{On the Markov approximation.}$
> > There are a few motivations behind our approximation. First, it is only possible to simulate fBM in discrete time naturally requiring some form of an approximation. This also applies to any computationally tractable inference framework for fBM. The reason why we specifically choose a Markov-approximation is computationally and theoretically rooted. From a practical perspective, our approximation leads to a simple implementation, where number of driving processes can be quite small. From a theoretical aspect, the extensive literature and the established theory of Markovian systems are essential to design a principled inference framework, benefiting from the tools of stochastic analysis. For example, without the Markov property, we cannot invoke Girsanov's change of measure theorem and therefore cannot derive our variational bound.
> >
> > We certainly agree with the reviewer that a deep investigation into the errors attained by our approximation is important. However, this is a non-trivial endeavor which we leave for a future study. There are two reasons: (i) while Tong et al. (2022) as well as Harms (2019), and Bayer & Breneis (2023) achieve some form of theoretical results regarding Type II, we are not aware of any studies concerning Type I, which is what we use in practice; (ii) even for the Type II, our particular choice of $\gamma$ variables and optimization for $\omega$ prevents us from directly inheriting the aforementioned analysis.
> >
> > Nevertheless, We do provide a strong empirical case for Type II in Figures 6 and 7 as well as the process plots in App. F.1. We hope that these empirical findings foster future research into the approximation we use.
> >
> > $\textbf{Score-based generative modeling.}$
> > Finally, we would like to point out that the reviewer makes an interesting connection to score-generative models. Once again, thanks to the Markov property, our SDE can in fact be time-reversed and power a $\textit{fractional diffusion model}$, which we also leave for a future study.

---

### Official Review · Reviewer_J9Kf · 2023-11-02

**Soundness:** 3 good
**Presentation:** 3 good
**Contribution:** 3 good
**Rating:** 8
**Confidence:** 4

**Summary:**

A variational framework for learning latent SDEs driven by fractional Brownian motion (fBM) is introduced. To efficiently infer the variational parameters, an approximation of fBM by Markov processes is utilized, which reduces the problem to inferring parameters of SDEs driven by (non-fractional) Brownian motion.
In applications, the drift, diffusion and a control term are implemented as neural networks. Empirically, the methods capability to recover a fractional Ornstein-Uhlenbeck bridge and the Hurst index are evaluated. The method is also applied to a video modelling task.

**Strengths:**

- This work addresses a difficult problem and proposes a novel solution. The inference of latent SDEs driven by Brownian motion is already challenging, in this work the setting is extended to fractional Brownian motion while avoiding limitations of prior work (Tong et al, 2022)
- The background section of the paper is well written and does a good job in condensing the involved theory of stochastic calculus (for fBM) into a short conference paper.
- While the method is evaluated on video prediction, it appears to be applicable to any type of sequential data.

**Weaknesses:**

From my point of view, there are 2 main weaknesses in this submission (for details, see below).

 1. The method and the experiments are insufficiently described, and I have some questions in this regard. However, I am convinced, that the manuscript can be updated to be much more clear.

 2. The empirical evaluation is of limited scope. Qualitatively, the method is evaluated on 2 toy problems (fOU & Hurst index); quantitatively on a single synthetic dataset (stochastic moving MNIST). For the latter, only two baseline models from 2018 and 2020 are compared to. In consequence, the usefulness of the method is not established. On the plus side, there are 3 ablations / further studies.

Minor weaknesses.
- The method appears to be inefficient, with a training time of 39 hours on an NVIDIA GeForce RTX 4090 for one model per model trained on stochastic moving MNIST.

- A detailed comparison with Tong et al. (2022) (who also learn approximations to fBM) is missing. So far, it is only stated that Tong et al. did not apply their model to video data and that it is completely different. A clear illustration of the conceptual differences and a comparison of the pros and cons of each approach would be appreciated (summary in main part + mathematical details in supplementary material).

# Summary
For me, this is a borderline submission. On the one hand, the proposed method is novel, significant and of theoretical interest. On the other hand, there are clarity issues, a weak empirical evaluation and no clear use case. Expecting the clarity issues to be resolved, I rate the submission as a marginally above the acceptance threshold, as the theoretical strengths outweigh the empirical flaws.

----
# Details on major weaknesses


## Point 1 (clarity).

**Regarding the method.** After reading the method and experiment section multiple times, I still have no idea, how to implement it. I am aware of the provided source code, nevertheless I found the paper to be insufficient in this regard.

What I got from section E and Figure 4 is that:
- First, there is an encoding step that returns $h$, a sequence of vectors/matrices over time. Somehow, these vectors are used to compute $\omega$. I have no idea how this $\omega$ is related to the optimal one from Prop. 5.
- $h$ is given to a temporal convolution layer that returns $g$ and this $g$ has as many 'frames' as the input and is used as input to the control function $u$.
- The control, drift and the diffusion function are implemented as neural networks.
- An SDE solution is numerically approximated with a Stratonovich–Milstein solver.
- $\omega$ is used in the decoding step, I do not understand why and how.
- Where do the approximation processes $Y$ enter. How are they parametrized, is $\gamma$ as in Prop 5? Are they integrated separately from $X$?
- The ELBO contains an expectation over sample paths. How many paths are sampled to estimate the mean?
- Fig. 6: Why do the samples from the prior always show a 4 and a 7? Does the prior depend on the observations?

**Regarding Moving MNIST.**
What precisely is the task / evaluation protocol in the experiment on the stoch. moving MNIST dataset? I did not see it specified, but from the overall description it appears that a sequence of 25 frames is given to the model and the task is to return the same 25 frames again (with the goal of learning a generative model).

## Point 2 (empirical evaluation)
- The method is not evaluated on real world data.
- Quantitatively, the method is only evaluated on one synthetic dataset (stoch. moving MNIST).
- While the method is motivated by "*Unfortunately, for many practical scenarios, BM falls short of capturing the full complexity and richness of the observed real data, which often contains long-range dependencies, rare events, and intricate temporal structures that cannot be faithfully represented by a Markovian process"*, the moving MNIST dataset is not of this kind. It is not long range (only 25 frames) and there is no correlated noise.
- The method is only compared to 2 baselines (SVG, SLRVP) on moving MNIST.
- Table 1 does not show standard deviations.

Overall, this would be a far stronger submission, if the experiments were more extensive. This includes:
- Evaluation on more task and datasets, and specifically on datasets were this method is expected to shine, i.e., in the presence of correlated noise. The pendulum dataset of [Becker et al., Factorized inference in high-dimensional deep feature space, ICML 2019] would be one example.
- Comparison with more baselines. In particular, more recent / state-of-the-art methods that do not model a differential equation and the fBM model by Tong et al.


----

There is a typo in Proposition 1: "Markov rocesses"

**Questions:**

**Extension to different tasks**

It seems that in its current form the method can only be applied when the model output should match the model input. This is because the control function $u$ at each time point depends on the evidence at that time point. How would you extend the method to forecasting or interpolation tasks, where this is not possible?

---

**Proposition 5**

If I understand correctly, then $\hat B$ depends on $\omega$ and $\gamma$ and the goal is to find $\omega$ such that the L2 error of the approximation is minimal. The minimum is achieved when $\omega$ solves $A\omega=b$.
Does this equation have a solution? How large is the approximation error? How is $\omega$ computed; is it required to first compute the entire matrix $A$?

---

**Runtime**

I wonder how the runtime compares to Tong et al., and to a latent ODE.
Does K=0 in Fig. 9 correspond to a latent SDE with non-fractional Brownian motion?

Is Fig.9 qualitatively the same, if training time (forward+backward) is measured instead of inference time (only forward)?

In the "Impact of K and the #parameters on inference time." experiment, it is stated that *"the run-time is still dominated by the size of the neural networks"*. Presumably, this refers to the networks that implement the drift, diffusion and control and not the encoder / decoder networks?

---

> ### Author Response · Authors · 2023-11-22
> **Author Response 1**
>
> We thank the reviewer for their diligent and detailed review, and all the feedback. We are delighted to hear that they find our method to be a novel solution to a difficult problem, and our paper to be well-written. We have now significantly improved the paper in the light of the suggestions of the reviewer.
>
> $\textbf{On clarity.}$ We thank the reviewer for the helpful indication of their understanding of Figure 4 and Section E. In the light of these, we have considerably improved the readability and clarity of our paper. We further address the questions below.
>
> The static content vector $w$, in the explanation of the video modelling framework, should not be confused with the weights $\omega$ (Greek letter omega) in the fBM approximation. The approximation processes $Y_k$ are integrated jointly with $X$. The state of the SDE process is thus constructed as in Eq. (14): $Z = (X, Y_1, \dots, Y_K)$. The values of $\gamma$ are described in our Appendix for all experiments. For the video models, we only take one sample path to calculate the ELBO. For the bridge and time dependent Hurst experiments we take $16$ or $32$ samples.
>
> $\textbf{On the prior and Moving MNIST.}$ In our video modelling experiments, both the prior model and posterior model are parameterized. The prior is as defined in Eq. (13). The posterior model is the prior model with the added control function $u(t)$, as defined in Eq. (16). Since only the control function receives information of the data sequence, the posterior is capable of reconstructing the sequence during training. When the model is trained, we use the prior as a generative model. Specifically, due to maximizing the ELBO, the prior will learn to model the dataset. For example, to condition on the first frames of a movie and generate predictions, we first use the posterior model on the given frames. At the time point where we start predicting, we use the prior. Practically, this simply means setting $u(t)=0$.
>
> The Peak Signal-to-Noise Ratio (PSNR) metric is calculated frame-wise, averaged over time, by sampling $100$ videos conditioned on the first $5$ frames. The best PSNR is selected from these $100$ samples.
> This approach is adapted from SLRVP to allow direct comparison.
> This metric is representing how good the model captures the data.
> Please note that we have considerably improved the manuscript under the 'Latent video models' section and incorporated this explanation.
>
> $\textbf{On empirical evaluations.}$
> Following the reviewer's suggestion, we added extra experiments on a $\textbf{real-world dataset composed of videos of the chaotic motion of a two-link double pendulum}$, released by IBM under https://developer.ibm.com/exchanges/data/all/double-pendulum-chaotic/ [1]. We thank the reviewer for this to-the-point suggestion. We provide the results below as well as in our revised Sec. 5:
>
> | Model |   ELBO  | PSNR  |
> |-------|:-------:|-------|
> | BM    | −545.13 | 26.11 |
> | fBM   | −636.61 | 27.09 |
>
> Even though this system is governed by ODEs, its chaotic nature poses a big challenge for any ODE-based method to faithfully capture its behaviour. Arguing that the hard-to-predict transitions can be modeled by stochastic noise, we evaluate our video model using both BM and fBM, where the latter attains a lower ELBO as a higher PSNR. We have $\textbf{revised Sec. 5}$ and incorporated qualitative results indicating the diversity of our samples. We hope that this new result clarifies some of the concerns and strengthens our empirical evaluation.
>
> $\textbf{Comparisons to Tong et al. (2022).}$ Note that, Tong et al. (2022) only reports Hurst index estimation in their experiments. Following the same protocol, we have already compared our method against theirs in time-dependent Hurst index. This experiment ($\textbf{Fig. 3}$) not only extends our method to $\textit{ multifractional Brownian Motion}$, but also shows that our approach yields more correct Hurst estimates, especially in the extremes. This is further supplemented by our conceptual discussion in $\textbf{App. E.2}$. An important difference is that our work covers Type I, whereas Tong et al. (2022) does not.
>
> $\textbf{Further applications.}$ Our experiments contain the applications of video prediction, which is an extrapolation task that predicts future frames given past frame. Interpolation is an easier problem and certainly possible within our framework. Note that, our bridge experiments already contain non-Markovian boundary value problems. Our versatile framework can be extended to other applications, which we leave for a future study.
>
> [1] Asseman, Alexis, Tomasz Kornuta, and Ahmet Ozcan. "Learning beyond simulated physics." (2018).

---

> > ### Author Response · Authors · 2023-11-22
> > **Author Response 2**
> >
> > $\textbf{Proposition 5.}$
> > We indeed compute the weights by solving $\mathbf{A}{\omega}=\mathbf{b}$ by first computing $\mathbf{A}$, which depends on the number of processes chosen, which determines the size of $\mathbf{A}$. We included a proof in the appendix that $\mathbf{A}$ is a positive definite matrix, thus that there is exactly one solution for $\omega$:
> > There is exactly one solution if $\mathbf{A}^{(I,II)}$ is positive definite, which is defined as
> > $$
> >     \omega^T \mathbf{A}^{(I,II)} \omega > 0 \text{ for all } \omega \in \mathbb{R}^K \setminus \{\mathbf{0}\} \, .
> > $$
> > Recall that (Eqs. 67-70 and 74-78)
> > $$
> >     \omega^T \mathbf{A}^{(I,II)} \omega = \int_0^T \mathbb{E}\left[{\hat{B}^{(I,II)}_H(t)^2}\right] \mathrm{d}t.
> > $$
> > Hence $\mathbf{A}^{(I,II)}$ positive definite (indicating a single solution) if:
> > $$
> >     \int_0^T \mathbb{E}\left[{\hat{B}^{(I,II)}_H(t)^2}\right] \mathrm{d}t > 0  \text{ for all } \omega \in \mathbb{R}^K \setminus \{\mathbf{0}\}
> > $$
> > Recall that $\hat{B}^{(I,II)}_H(t)$ is a linear combination of $K$ Ornstein-Uhlenbeck processes with speed of mean reversion $\gamma_k$ driven by the same Brownian motion.
> > Under the trivial condition that $\gamma_i \neq \gamma_j$ (so they can not cancel out) and $\gamma_k < \infty$, this will never be $0$.
> > Hence the last inequality holds unless $\hat{B}^{(I,II)}_H(t) = 0$. This concludes the proof.
> >
> > In practice, we never explicitly invert $\mathbf{A}$ but use JAX's linear symmetric system solver (https://jax.readthedocs.io/en/latest/_autosummary/jax.scipy.linalg.solve.html).
> > This is faster and more numerically stable than inverting $\mathbf{A}$.
> >
> > The approximation error is presented in Figs. 6 (analytically) and 7 (for sampled paths). Please see also $\textbf{App. F.1}$ for examples of generated approximated fBM paths vs. true fBM paths for various Hurst indices.
> >
> > $\textbf{Runtime.}$ We thank the reviewer for this suggestion. While there are implementation differences, we still find it important to analyze how our runtime compares to Tong et al. We do so on the task of time dependant Hurst estimation and report the runtime for a single forward and backward pass below:
> >
> > |                    |   Forward Pass   |   Backward Pass  |
> > |--------------------|:----------------:|:----------------:|
> > | Tong et al. (2022) | 396 ms ± 29.7 ms | 461 ms ± 24.1 ms |
> > | Ours               | 28.7 ms ± 350 µs | 741 ms ± 20.3 ms |
> >
> > While Tong et al. (2022) has a slight advantage regarding the backward pass, we are an order of magnitude faster when it comes to inference. We must admit that implementation differences also play a role here. Our method is implemented in JAX whereas we use the official PyTorch implementation of Tong et al.
> >
> > Fig. 9 only contains values for $K>3$. We have clarified this by starting the $x$-axis at $3$. We thank the reviewer for the suggestion of incorporating the backward pass as well. We have re-run this experiment including the backward pass and results are qualitatively the same. We have updated that Figure in our manuscript.
> >
> > The reviewer is right that for this experiment we focused on the drift, diffusion and control function neural networks.
> >
> > We hope that we have addressed all the major concerns of the reviewer. We have incorporated these into our manuscript.

---

> > ### Comment · Reviewer_J9Kf · 2023-11-22
> > **A brief follow up**
> >
> > 1. **Regarding the prior process.** Did I understand you correctly, that the prior SDE (parameters) itself does not depend on the observations, but the initial state (or the state from which you start predicting) does?
> > 2. **Regarding comparison with Tong et al.** You refer to a conceptional discussion in **App E.2.**. Could you be more precise, as I did not find this discussion in the section E.2 (titled time dependent hurst index).

---

> > > ### Author Response · Authors · 2023-11-23
> > >
> > > 1. The reviewer is right that the initial state is sampled from the learned posterior distribution $q_{x_1}$, which depends on the data. This is a common methodology in training neural SDEs (see for example Li et al. (2020)). We explain this in more detail in **App. E.3**.
> > > 2. We have now moved this conceptual discussion to the manuscript (revised) under **App. A**. We kindly refer the reviewer to our second response to Reviewer **9F2a** where we also discuss this topic. We thank the reviewer for this correction.

---

### Meta-Review · Area_Chair_K8PK · 2023-12-05

**Metareview:**

This is a topical paper that addresses the difficult problem of inference of latent SDEs driven by fractional Brownian motion while avoiding limitations present in previous work. The reviewers appreciated the problem setup, the methodological contributions, and the condensed presentation of the rather technical topic. On the other hand, the paper could have provided a more through evaluation/comparison setup in the experiments.

This paper should be of interest to the ICLR community and is a good fit to the conference program.

**Justification For Why Not Higher Score:**

The topic is rather specialised and technical and might not be interesting for a wide-enough audience for an oral.

**Justification For Why Not Lower Score:**

This paper could also just be a poster, but could also be interesting as spotlight due to its technical nature.

---

### Decision · Program_Chairs · 2024-01-16

Accept (spotlight)